# FROM INTERNAL REPRESENTATIONS TO TEXT QUALITY: A GEOMETRIC APPROACH TO LLM EVALUATION

## ABSTRACT

This paper bridges internal and external analysis approaches to large language models (LLMs) by demonstrating that geometric properties of internal model representations serve as reliable proxies for evaluating generated text quality. We validate a set of metrics—including Maximum Explainable Variance, Effective Rank, Intrinsic Dimensionality, MAUVE score, and Schatten Norms measured across different layers of LLMs, demonstrating that Intrinsic Dimensionality and Effective Rank can serve as universal assessments of text naturalness and quality. Our key finding reveals that different models consistently rank text from various sources in the same order based on these geometric properties, indicating that these metrics reflect inherent text characteristics rather than model-specific artifacts. This allows a reference-free text quality evaluation that does not require human-annotated datasets, offering practical advantages for automated evaluation pipelines.

## 1 INTRODUCTION

The rapid advancement of large language models (LLMs) has necessitated the development of methods for analyzing their internal mechanisms and the properties of generated text. Approaches to studying the geometric properties of representations in language models can be broadly categorized into two categories: **internal** or mechanistic methods, which investigate the model's intermediate representations, and **external** methods, which analyze the properties of text embeddings captured via some embedding model. Internal evaluation mainly considers **model properties** within which, these measures were made (Yin et al., 2024; Viswanathan et al., 2025; Roy & Vetterli, 2007), while external measures mainly focus on evaluation of **text properties** given text embeddings (Zhao et al., 2019; Tulchinskii et al., 2023; Kuznetsov et al., 2024).

In this work, we bridge these two perspectives by demonstrating that internal geometric metrics can serve as powerful proxies for evaluating model performance and text quality. We show that properties intrinsic to the model's representations correlate strongly with established external evaluation metrics, providing a novel framework for model assessment that does not require human-annotated datasets.

Here, we illustrate this idea through the following setup. Let us consider a set $\mathcal{T}$ of tester models to analyze text generated by a separate set $\mathcal{G}$ of generator models. We empirically demonstrate that the internal representations of any model within $\mathcal{T}$ can be used to consistently rank the quality of text produced by models in $\mathcal{G}$. Crucially, all tester models in $\mathcal{T}$ yield the same ranking of generators in $\mathcal{G}$ (see Figure 1), indicating that these geometric metrics capture intrinsic properties of the text itself, independent of the specific model used for analysis. All utilized models are described in Section 3.1.

Conventional LLM evaluation heavily relies on annotated data and accuracy-based metrics (Ni et al., 2025; Hu & Zhou, 2024). This approach has several disadvantages. First, it requires the development of a new annotated benchmark for each application scenario, which is impractical for rapid development because evaluating models on many such datasets is time-consuming. Second, most benchmarks prioritize utility measuring how accurate a model is (Ni et al., 2025) over generated text quality. This raises a long-standing question, reminiscent of Kant's "Beauty vs. Utility" dilemma (Clewis, 2018). A text containing valuable information, but that is unpleasant to read may be as useless as a text with no valuable information at all, since it causes discomfort to read (Radivojevic et al., 2024).

This brings us to a point where our results find their practical use since we need a way to evaluate text quality without resorting to annotation and labeling. Finally, we have not yet discussed how we define **text quality**. While the concept may seem ambiguous, previous researchers have defined it as text naturalness (Hassan et al., 2024; Xia et al., 2024; Sen et al., 2025) - that is, text that closely resembles human language. Building on this, we evaluate and find that some intrinsic measures correlate with established metrics for measuring text naturalness. That means that we can use a small proxy-tester model to assess the naturalness of generated text via its intermediate representations.

**Our specific contributions are as follows:**

- We validate a set of metrics: Maximum Explainable Variance, Effective Rank, Intrinsic Dimensionality, MAUVE score and Schatten Norms that measure internal representations across different layers of six LLMs and demonstrate strong correlations for certain metrics.

- Using these metrics, we show that all six tester models consistently rank text generated by eight generator models in the same order, indicating that these metrics reflect properties of the text itself rather than characteristics of the specific model used for analysis. For our tester models, we use models of different sizes from 0.5B to 8B. Moreover, we evaluated intrinsic properties of diffusion-based LLM and find it ranks generated texts as good as autoregressive models.

- We establish correlations between our geometric metrics and present in the literature six language "naturalness" measures.

- We propose Intrinsic Dimensionality and Effective Rank as universal reference-free measure of text quality based on fundamental geometric properties, offering an alternative to reference-based metrics.

## 2 RELATED WORK

**Geometric properties** such as Intrinsic Dimension (ID) applied to text embeddings exhibit varying values for different languages, and ID is consistently lower for AI-generated text compared to human-written content (Tulchinskii et al., 2023). By applying sparse autoencoders to LLMs residual streams, (Kuznetsov et al., 2025) has demonstrated that LLMs produce text with distinctive stylistic features and that text generated by different model classes can be distinguished from one another. Viswanathan et al. (2025) showed that ID is higher in middle layers of LLMs compared to other layers, and that ID for randomized text is significantly higher—indicating models struggle to compress it. Intrinsic dimensionality across different layers has been considered as a measure of generalization capability in both convolutional networks (Ansuini et al., 2019) and LLMs (Roy & Vetterli, 2007). Godey et al. (2024) demonstrated that anisotropy is an inherent property of transformer-based models and can itself serve as a model characteristic, potentially useful for detecting hallucinations (Yin et al., 2024).

**The evaluation of text naturalness and quality** typically employs various metrics. GPT-2 Perplexity is commonly used to assess the fluency and naturalness of generated text by measuring how well a pre-trained model predicts the text sequence (Chang et al., 2024). Traditional metrics like BLEU and ROUGE measure lexical similarity to human-written references, though they struggle to capture nuanced semantic aspects (Wang et al., 2023). At the same time, BLEURT (Yan et al., 2023) demonstrates higher correlation with human judgments of overall text quality—capturing semantic adequacy and entailment, along with aspects of fluency and grammar. MAUVE has emerged as a prominent method for quantifying similarity between neural-generated and human-written text by computing divergence curves between their distributions, with higher scores indicating more coherent, human-like text (Xia et al., 2024; Sen et al., 2025). Other approaches include using compression ratios with algorithms like gzip to estimate text diversity (Chang et al., 2024) and examining statistical differences in linguistic features between human and LLM-generated texts, including average subtree height, dependency tree height, and sentence length (Yu-Te Lee et al., 2024).

Table 1: Rankings of text generators across studied metrics. Higher rank presents better performance, values are aggregated among all tester models $\mathcal{T}$. Metrics definitions are provided in Table 2.

| Text origin | Schatten | MEV | ERank | Resultant | MAUVE | MLE | MOM | MADA | CorrInt | Average |
|---|---|---|---|---|---|---|---|---|---|---|
| Starling-lm-7b-beta | 8.0 | 9.0 | 9.0 | 2.0 | 6.0 | 9.0 | 7.0 | 3.0 | 8.0 | 9.0 |
| **Human text** | 9.0 | 8.0 | 8.0 | 1.0 | 7.0 | 2.0 | 8.0 | 5.0 | 9.0 | 8.0 |
| Deepseek-R1 | 5.0 | 4.0 | 4.0 | 6.0 | 4.0 | 7.0 | 9.0 | 8.0 | 7.0 | 7.0 |
| LLama-3.1-8B-it | 7.0 | 7.0 | 6.0 | 3.0 | 1.0 | 8.0 | 2.0 | 9.0 | 5.0 | 6.0 |
| Phi-3-medium-4k-it | 6.0 | 3.0 | 3.0 | 8.0 | 8.0 | 3.0 | 6.0 | 2.0 | 3.0 | 5.0 |
| Gemma-2b-it | 4.0 | 6.0 | 7.0 | 4.0 | 5.0 | 4.0 | 4.0 | 1.0 | 6.0 | 3.0 |
| Mistral-7b-it | 3.0 | 5.0 | 5.0 | 5.0 | 3.0 | 5.0 | 5.0 | 6.0 | 4.0 | 3.0 |
| Phi-3-mini-128k-it | 1.0 | 1.0 | 1.0 | 9.0 | 9.0 | 1.0 | 1.0 | 7.0 | 1.0 | 2.0 |
| Qwen-2.5-7b-it | 2.0 | 2.0 | 2.0 | 7.0 | 2.0 | 6.0 | 3.0 | 4.0 | 2.0 | 1.0 |

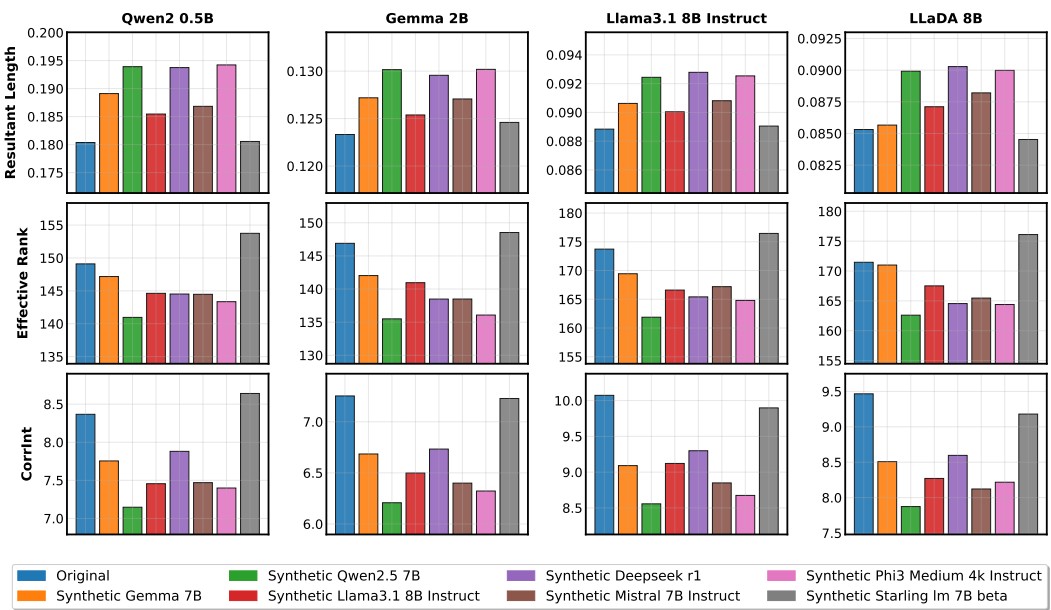

Figure 1: The ranking of eight generators $\mathcal{G}$ via four tester models $\mathcal{T}$ with different sizes from 0.5B to 8B (Qwen2 0.5B, Gemma 2B, Llama3.1 8B Instruct and diffusion LLaDA 8B) and three geometric metrics (Resultant Length, Effective Rank and CorrInt). As could be seen, the rankings of the generators models are similar.

## 3 METHODS

### 3.1 MODELS

For our experiments, we utilized six tester models ($\mathcal{T}$) to evaluate text from eight generator models ($\mathcal{G}$). **Tester Models** ($\mathcal{T}$): Gemma-1-7b, Gemma-2b-it, LLama-3.1-8B-it, Qwen-2.5-7b-it, Qwen-2-0.5B, LLaDa-8B. **Generator Models** ($\mathcal{G}$): Gemma-2b-it, Qwen-2.5-7b-it, LLama-3.1-8B-it, Deepseek-R1, Mistral-7b-it, Phi-3-medium-4k-it, Phi-3-mini-128k-it, Starling-lm-7b-beta.

### 3.2 DATASETS

To evaluate the naturalness of generated text, certain metrics require reference texts. To this end, we created pairs of original and rewritten reviews by prompting generator models to paraphrase movie reviews while preserving their meaning and approximate length.

**Prompt:** *"Rewrite this text in a different style while preserving the main idea. Try to maintain the original length and language. Output only the rewritten text. Original text:"*

We conducted our evaluation across three languages: English, German, and Russian. For English, we used the IMDB dataset (Maas et al., 2011), for Russian, we used Kinopoisk movie reviews from Kaggle (Klemin, 2024), and for German, we used movie reviews from (Guhr et al., 2020). From each dataset, we sampled 1,000 reviews. With eight generator models, this resulted in a total of 8,000 text pairs (1,000 reviews × 8 models) for each language.

### 3.3 METRICS FOR TEXT NATURALNESS AND QUALITY

We employ three reference-based and four-reference free metrics from the literature, as discussed in Section 2. For reference-based metrics, we use ROUGE, BLEURT and MAUVE. These metrics compare the original text and its rewritten version, as described in Section 3.2. For reference free metrics we employ Compression Rate (CR) with ZIP compressor, GPT perplexity and lexical features such as average length and the standard deviation of length.

### 3.4 MEASURING GEOMETRIC PROPERTIES

Let the matrix $X_g^{(l)} \in \mathbb{R}^{n \times d}$ denote the hidden state of the layer $l$ of the tester model $t \in \mathcal{T}$. We obtain $X^{(l)}$ from MLP blocks, after activation functions and before residual connection. The $X_g$ corresponds to text generated by generator model $g \in \mathcal{G}$. Here $n$ is the length of the input text and $d$ is the hidden dimension. For some metric $\mathbf{R} : \mathbb{R}^{n \times d} \to \mathbb{R}$, the layer wise scores $s_L^R$ and the average score $s^R$ for $t_i$ with $L$ layers are defined as:

$$s_L^R(X_g) = [R(X_g^{(1)}), R(X_g^{(2)}), ..., R(X_g^{(L)})] \in \mathbb{R}^L,$$

$$s^R(X_g) = \frac{1}{L} \sum_{l=1}^{L} R(X_g^{(l)}) \in \mathbb{R}, \tag{1}$$

where $X = (X^{(1)}, X^{(2)}, ..., X^{(L)})$ is the set of hidden states of each layer of tester model $t_i$. Finally, we treat $s^R(X_g)$ as a measure for text generated by model $g$, this allows us to compare text generated by various models in $\mathcal{G}$ with each other.

### 3.5 A SET OF METRICS - $\mathbf{R}$

Until now, we have not formally defined the set of metrics $\mathbf{R}$ used in this work. In this section, we provide precise mathematical formulations and categorize them into three conceptual groups based on what aspect of text representations they capture: Representational Magnitude, Diversity, and Naturalness.

Table 2: Summary of evaluation metrics used in this work. Direction indicates whether higher (↑) or lower (↓) values are generally preferable for natural, diverse text.

| Metric | Category | Direction | Brief Description |
|---|---|---|---|
| Schatten Norm | Representational Magnitude | — | Global spectral energy, measures magnitude/stability. |
| MEV | Diversity | ↓ | Fraction of variance in top singular values. |
| Resultant Length ($R$) | Diversity | ↓ | Bigger value means all vectors point into the same direction. |
| Effective Rank (ERank) | Diversity | ↑ | Entropy of normalized singular values. |
| MAUVE | Naturalness | ↑ | Distributional alignment to human text in feature space. |
| Intrinsic Dimension (ID) | Naturalness | ↓* | Estimated manifold dimension — *lower = more predictable. |

The **Maximum Explainable Variance (MEV)** (Razzhigaev et al., 2024) and **Resultant Length** (Ethayarajh, 2019) — often collectively referred to as measures of **Anisotropy** — quantify the degree of directional concentration in token embeddings. Lower values indicate more isotropic, evenly distributed representations, which are associated with greater semantic diversity and complexity. Empirically, generated text tends to exhibit higher MEV and Resultant Length compared to human text, making these metrics useful for distinguishing synthetic from natural content.

Let $X^{(l)} \in \mathbb{R}^{N \times d}$ denote the matrix of token representations from layer $l$, where each row corresponds to a token embedding. Let its singular value decomposition be:

$$X^{(l)} = U\Sigma V^\top, \quad \Sigma = \mathrm{diag}(\sigma_1, \sigma_2, \ldots, \sigma_r), \quad r = \min(N, d), \tag{2}$$

where $\sigma_1 \geq \sigma_2 \geq \cdots \geq \sigma_r \geq 0$ are the singular values.

**Maximum Explainable Variance (MEV).**

$$\text{MEV}(X^{(l)}) = \frac{\sigma_1^2}{\sum_{i=1}^{r} \sigma_i^2}. \tag{3}$$

This measures the proportion of total variance explained by the first principal component. High MEV indicates that representations are dominated by a single direction — a signature of representational collapse or anisotropy.

**Resultant Length** ($R$).  Let $\mathbf{x}_i = \frac{X_{i,:}^{(l)}}{\|X_{i,:}^{(l)}\|_2}$ be the unit-normalized $i$-th token embedding. Then:

$$R(X^{(l)}) = \left\| \frac{1}{N} \sum_{i=1}^{N} \mathbf{x}_i \right\|_2. \tag{4}$$

This measures the magnitude of the mean direction vector. $R = 0$ implies perfect isotropy; $R = 1$ implies perfect alignment. Like MEV, it is typically elevated in generated text.

**Schatten Norm.**  The **Schatten-$p$ norm** (Bhatia, 1997) provides a family of matrix norms that quantify global spectral energy in the representation matrix $X^{(l)}$. For $p \geq 1$, it is defined as:

$$\|X^{(l)}\|_{S_p} = \left( \sum_{i=1}^{r} \sigma_i^p \right)^{1/p}, \tag{5}$$

where $\sigma_i$ are the singular values from the SVD in equation 2. The Schatten norm is not inherently better when higher or lower, its interpretation depends on the application. For instance, smaller nuclear norm may indicate more compact representations, while larger Frobenius norm may reflect higher activation energy or scale. It is often used for regularization, stability analysis, or comparing representation magnitudes across models or layers.

In contrast to the above, the **Effective Rank** Roy & Vetterli (2007) quantifies the *effective dimensionality* or *diversity* of the representation space. It is a continuous, entropy-based approximation of matrix rank, robust to small perturbations.

Let $p_k = \frac{\sigma_k}{\sum_{i=1}^{r} \sigma_i}$ be variance proportion of the $k$-th singular value. Then:

$$\text{ERank}(X^{(l)}) = \exp \left( - \sum_{k=1}^{r} p_k \log p_k \right). \tag{6}$$

High ERank indicates that the model utilizes many orthogonal directions to encode information, which can be seen as a sign of diverse representations.

**MAUVE** (Pillutla et al., 2021) measures the divergence between generated and human text by comparing their distributions in a quantized embedding space. Let $P$ and $Q$ denote these distributions, quantized into a discrete space via clustering. Then:

$$\text{MAUVE}(P, Q) = 1 - \inf_{\lambda \in [0,1]} [\lambda \cdot D_{\text{KL}}(P' \| R_\lambda) + (1 - \lambda) \cdot D_{\text{KL}}(Q' \| R_\lambda)], \tag{7}$$

where $R_\lambda = \lambda P' + (1 - \lambda)Q'$. MAUVE produces a score between 0 and 1, where higher values indicate better alignment with human text. While designed for text, MAUVE can be applied to other modalities by using domain-specific embeddings and is best used to compare different models or decoding strategies rather than for absolute evaluation. We analyze texts and the internal representations $X_g^{(l)}$ mentioned above based on approach proposed in the original paper.

**Intrinsic Dimensionality (ID).**  Intrinsic Dimensionality estimates the manifold dimension on which text representations lie. The idea of ID is to measure the minimum number of parameters required to represent the data without significant loss of information, which could be measured both locally and globally. Common estimators include the **Correlation Dimension** (Grassberger & Procaccia, 1983), the **Maximum Likelihood Estimator (MLE)** (Levina & Bickel, 2004) and others Farahmand et al. (2007); Amsaleg et al. (2018). While not directly based on log-likelihood, lower ID often correlates with higher predictability, suggesting that the text lies on a simpler, more structured manifold. This allows to consider ID as an indicator of the model's ability to generalize.

# 4    RESULTS

We present our results in the following order:

1.  We demonstrate that different $\mathcal{T}$ models yield identical rankings of texts generated by various $\mathcal{G}$ models, based on the metrics **R**.

2.  We observe that certain geometric properties are strongly correlated with one another, while others are not, and we offer a brief interpretation of these relationships.

3.  We analyze whether these observations hold across different languages, specifically, Russian and German.

4.  Finally, we show that some geometric metrics exhibit strong correlations with the text quality metrics introduced in Section 3.3.

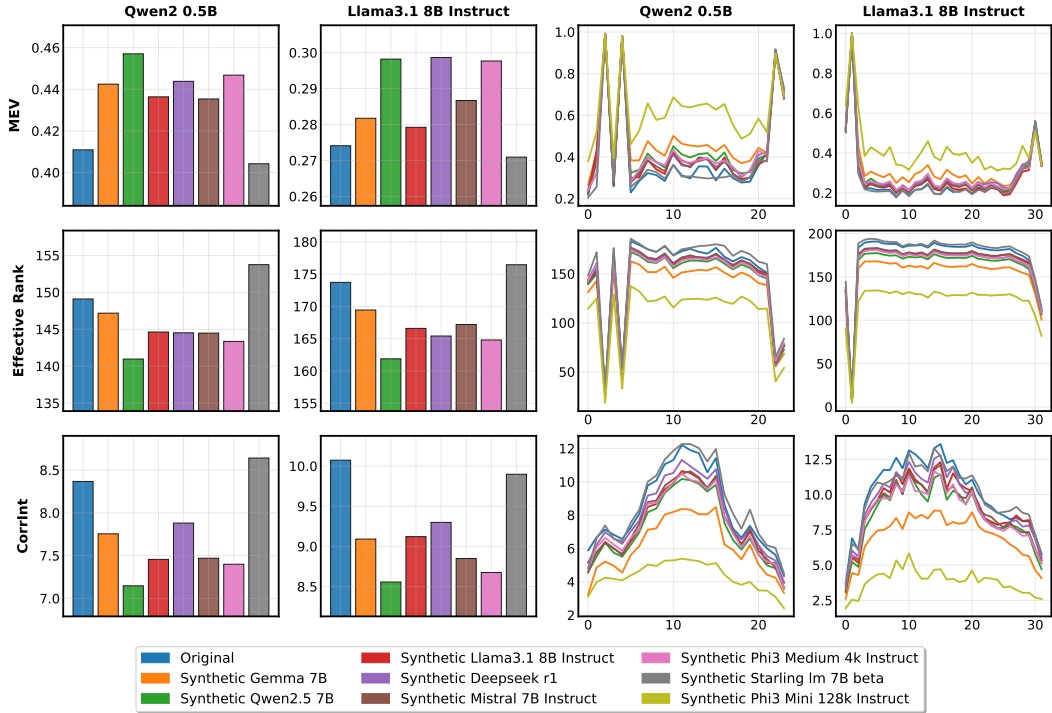

Figure 2: The average across layers (left two columns) and layer-wise (right two columns) metrics: MEV, Effective Rank and CorrInt for with text generated by various models models $\mathcal{G}$ and tester models $\mathcal{T}$: Qwen2 0.5B and Llama3.1 8B Instruct. Original means human written text, while all generated text simply represents rewritten via LLMs original text, which preserves semantic meaning.

## 4.1    CONSISTENT RANKING

As shown in the right panel of Figure 2, different models produce consistent rankings for texts generated by various generators at corresponding layers. Additional layer-wise results are provided in Appendix Figures 8 and 9. The left panel of Figure 1 and Figure 2 further illustrate that average **R** scores across layers—computed as in Equation 1, consistently rank the generated texts.

To assess these results quantitatively, we compute Spearman correlations among all $\mathcal{T}$ models using their **R** scores. As shown in Appendix Figure 17, the minimum correlation is $0.947$ (between the diffusion-based LLaDa-8B and Gemma-1-2B), with all other correlations exceeding this value and remaining statistically significant.

This strong agreement enables the use of small, arbitrary tester models to a highly practical advantage in real-world applications. Our aggregated ranking across all tester models presented in

Table 1 identifies **Starling-LM-7B** as the most "natural-sounding" model, **Deepseek-R1** as the second place, and **Phi3-mini-128k-it** as the least natural-sounding.

Notably, tester models $\mathcal{T}$ vary significantly in size (from 0.5B to 8B parameters), architecture, and training paradigm—including Qwen2-0.5B, Llama-3.1-8B-Instruct, and LLaDA-8B—yet they produce highly consistent rankings of generator models $\mathcal{G}$ (see Figures 1 and 2). Remarkably, even newer diffusion-based language models like LLaDA-8B Nie et al. (2025) separate synthetic texts similarly to autoregressive LLMs.

While all metrics consistently rank generated texts, an important question remains: *Which of these metrics best correspond to actual model quality?* We address this in Section 4.4.

## 4.2 Analysis of Geometric Scores and Interpretations

Figure 3 presents pairwise correlations between **R** scores. We observe that **Intrinsic Dimensionality metrics** are most strongly correlated with each other. Besides, we see a strong **positive correlation** between ERank (Equation 6) and CorrInt, which we hypothesize both reflect representation diversity. A strong **negative correlation** between two groups, (MEV and Resultant Length) vs. (CorrInt and ERank), suggests that MEV and Resultant Length may indicate less diverse or more anisotropic representations. The **Schatten norm** is the least correlated metric overall, suggesting it may be least connected to text quality or diversity. However, an interesting pattern emerges in Figures 8 and 9: the Schatten norm increases almost monotonically across layers, likely due to accumulated residual connections.

As seen in Figures 1 and 2, metrics such as Effective Rank (Equation 6), Schatten Norm (Equation 5), and Intrinsic Dimensionality are consistently higher for original human-written text than for synthetic text. This can be explained by the greater complexity and unpredictability of natural language, which leads to higher-rank, more isotropic embeddings. Consequently, better generator models producing less predictable, more human-like text, tend to yield higher values for these metrics.

Conversely, metrics like MEV, Resultant Length, and MAUVE are lower for original text, reflecting its lower anisotropy compared to synthetic outputs Razzhigaev et al. (2023). For comprehensive comparisons across models and metrics, see Figures 6 and 7 (average metric comparison) and Figures 8 and 9 (layer-wise analysis) in Appendix A.

## 4.3 Other than English languages

In this section, we discuss whether our findings hold true for other than English languages, German and Russian. The results in Figure 5 (and more in Appendix Figure 10) reveal an interesting pattern. On average, most of the metrics are "better" for English language but if we compare original German or Russian texts with generated German or Russian texts we would observe some gap. However, the gap in metrics between original and generated texts for non-English texts is much smaller, thus validity of text quality evaluation for non-English text with proposed approach requires additional investigation.

## 4.4 Text quality assessment with geometric metrics

Table 3: Comparison of text quality or naturalness metrics across original and generated texts.

| Model | CR | ROUGE-L ($\uparrow$) | BLEURT ($\uparrow$) | GPT-PPL ($\downarrow$) | MAUVE ($\uparrow$) | Avg Len | Std Len |
|---|---|---|---|---|---|---|---|
| Original | 0.5754 | — | 0.0000 | 42.35 | — | 18.25 | 7.03 |
| Gemma-2b-it | 0.5957 | 0.3534 | -0.1764 | 31.10 | 0.01 | 17.33 | 3.26 |
| Qwen-2.5-7b-it | 0.6148 | 0.4438 | -0.1499 | 38.51 | 0.06 | 16.18 | 3.71 |
| LLama-3.1-8B-it | 0.6072 | 0.3028 | -0.3047 | 27.24 | 0.03 | 17.93 | 3.70 |
| Deepseek-R1 | 0.6093 | 0.4286 | -0.1108 | 56.08 | 0.15 | 17.05 | 11.51 |
| Mistral-7b-it | 0.5903 | 0.3712 | -0.1663 | 35.57 | 0.11 | 16.75 | 3.70 |
| Phi-3-medium-4k-it | 0.6018 | 0.3304 | -0.2599 | 47.28 | 0.11 | 17.32 | 4.08 |
| Phi-3-mini-128k-it | 0.6182 | 0.2103 | -0.3561 | 49.32 | 0.02 | 22.22 | 5.13 |
| Starling-lm-7b-beta | 0.5791 | 0.4035 | -0.0904 | 30.47 | 0.01 | 19.22 | 3.68 |

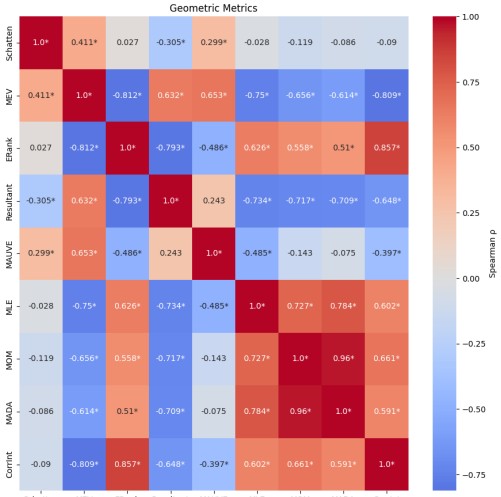
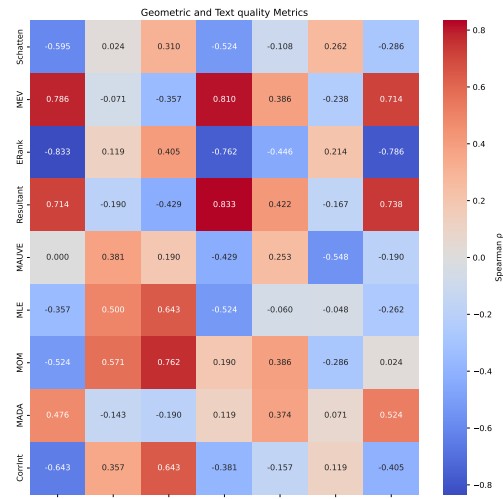

Figure 3: This Spearman correlation demonstrates similarity among different geometric **R** scores in terms of ranking texts generated by various models. Results are aggregated across both tester and generator models. Asterisk indicates FDR-corrected p-value $\leq 0.05$.

Figure 4: Here we demonstrate Spearman correlation between geometric **R** scores and text quality metrics. Results are aggregated across both tester and generator models.

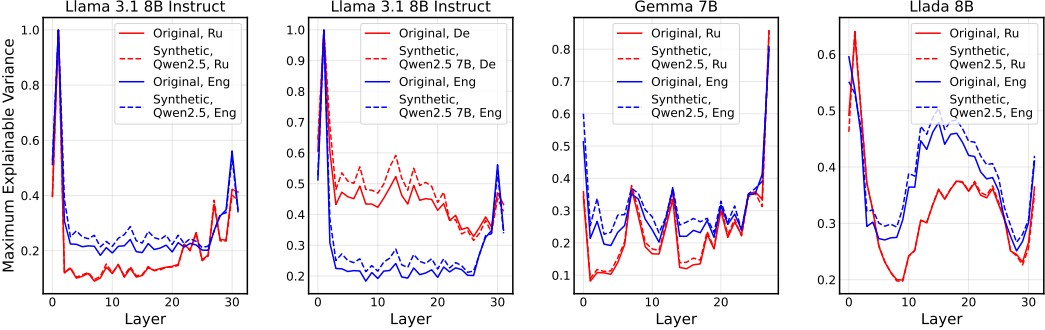

Figure 5: The comparison of Original and generated by Qwen2.5 7B synthetic texts on Russian (Ru), German (De) and English (Eng) for Maximum Explainable Variance 3 and various tester models $\mathcal{T}$.

The results in Table 3 reveal systematic differences between original human-written text and text generated by various LLMs across multiple quality and diversity metrics.

CR - compression ratio, which reflects textual redundancy, is lowest for the original text (0.575), indicating higher complexity and less repetition. Most models produce more compressible text, with Phi-3-mini-128k-it showing the highest ratio (0.618), suggesting lower lexical diversity.

ROUGE-L and BLEURT, which measure semantic similarity to reference texts, show that Gen-Qwen and Deepseek-R1 achieve the highest similarity scores, while Phi-3-mini-128k-it performs weakest — consistent with its higher compression ratio and lower coherence.

GPT Perplexity (GPT-PPL) is lowest for LLama-3.1-8B-it (27.24), indicating better fluency or alignment with GPT's expectations, while Deepseek-R1 has the highest perplexity (56.08), suggesting greater divergence from typical LLM output patterns.

MAUVE scores, which quantify distributional alignment between human and model-generated text, are generally low across all models ($\leq 0.15$), with Deepseek-R1 achieving the highest (0.15), suggesting it best approximates the human text distribution among generators.

Lexical features show that Phi-3-mini-128k-it generates significantly longer average outputs (22.22 tokens) with higher length variability (std = 5.13), potentially indicating verbosity or inconsistency. In contrast, Starling-LM-7B and Original text show more moderate, stable lengths.

Overall, no single model dominates across all metrics, but *Deepseek-R1 and Starling-LM-7B emerge as strong performers in semantic similarity and distributional fidelity*, while Phi-3-Mini shows signs of lower quality in multiple dimensions. These results align with model ranking obtained via geometric measures in Section 4.2.

Finally, to analyze more quantitatively how text-metrics align with geometric metrics, we demonstrate Spearman correlation in Figure 4. To provide correlations we used averaged scores among tester models and results obtained in Table 3, therefore, we do not report $p$ value here since more observations would be required. Spearman correlations reveal that geometric representation scores meaningfully align with text quality metrics. **ERank** (diversity) correlates negatively with GPT-PPL ($\rho = -0.76$) and positively with BLEURT ($\rho = 0.40$) suggesting isotropic representations associate with fluent, semantically coherent outputs. **MEV** and **Resultant Length** (anisotropy) correlate positively with GPT-PPL ($\rho = 0.81, 0.83$) and length variability indicating anisotropic representations link to less fluent, more unstable generation. **MOM** and **Corrint** best correlate with ROUGE-L and BLEURT (up to $\rho = 0.76$), capturing semantic alignment. **MAUVE** shows weak correlations ($|\rho| < 0.45$), suggesting it measures distributional fidelity orthogonal to representation geometry. **Schatten norm** correlates weakly with quality metrics, likely reflecting architectural effects (e.g., residual scaling) rather than output quality. These results support using **ERank**, **MEV** and **Corrint** as efficient, reference-free proxies for generation quality.

## 5 LIMITATIONS

The current study has several limitations that warrant discussion. Firstly, while our geometric metrics show strong consistency across diverse tester models, their absolute reliability as universal proxies for text quality, especially across a wider array of languages and domains beyond English, German, and Russian movie reviews, requires further validation. The observed smaller performance gap between human and synthetic text for non-English languages suggests potential cultural or linguistic biases in the metrics or the underlying models.

Secondly, our correlation analysis between geometric and traditional text quality metrics is based on aggregated scores from a relatively small set of generator models. This aggregation, while demonstrating clear trends, limits the statistical power for calculating precise p-values and may mask model-specific nuances.

## 6 CONCLUSION

This work establishes a robust and practical framework for evaluating the quality of text generated by large language models (LLMs) through the geometric analysis of their internal representations. Our key finding is that fundamental geometric properties, specifically Intrinsic Dimensionality, Effective Rank, and Maximum Explainable Variance—serve as reliable, reference-free proxies for text naturalness and quality. Crucially, these metrics produce consistent rankings of text generators across a diverse set of tester models, ranging from 0.5B to 8B parameters and encompassing both autoregressive and diffusion-based architectures. This consistency demonstrates that the metrics capture intrinsic properties of the text itself, rather than artifacts specific to any single evaluator model.

The strong correlations observed between these geometric measures and established external metrics for text quality (such as BLEURT and GPT-2 perplexity) validate their practical utility. For instance, higher Effective Rank and lower Maximum Explainable Variance are consistently associated with more human-like, diverse, and semantically coherent text. This allows practitioners to leverage small, efficient tester models for rapid, automated quality assessment without the need for expensive human annotations or reference texts.

**Acknowledgment on LLM assisted writing:** This paper used open access Qwen3-Max, in some parts of the paper, for proofreading and text rephrasing in accordance with formal style.

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

# A   APPENDIX

In the following appendix sections, we present additional experiments that include various metrics $\mathbf{R}$, tester models $\mathcal{T}$, and generator models $\mathcal{G}$. The metrics we utilize as $\mathbf{R}$ are Schatten Norms, Maximum Explainable Variance, Effective Rank, Resultant Length, MAUVE, and Intrinsic Dimensionality metrics (MLE, MOM, MADA, and CorrInt).

For the tester models $\mathcal{T}$ we use Gemma-1-7b, Gemma-2b-it, LLama-3.1-8B-it, Qwen-2.5-7b-it, Qwen-2-0.5B and LLaDa-8B.

For the generator models $\mathcal{G}$ we utilize Gemma-2b-it, Qwen-2.5-7b-it, LLama-3.1-8B-it, Deepseek-R1, Mistral-7b-it, Phi-3-medium-4k-it, Phi-3-mini-128k-it and Starling-lm-7b-beta.

The structure of the section is:

- The average metrics for different tester $\mathcal{T}$ and generator $\mathcal{G}$ models are demonstrated in Figures 6 and 7.
- For the metrics for different layers of tester $\mathcal{T}$ models for various generator $\mathcal{G}$ models see Figures 8 and 9.
- The layerwise metrics for Russian and English languages and different tester and generator models are demonstrated in Figures 10, 11, 13, 14, 15 and 16.
- The Spearman correlation among all $\mathcal{T}$ models for all geometric $\mathbf{R}$ scores (Figure 17).

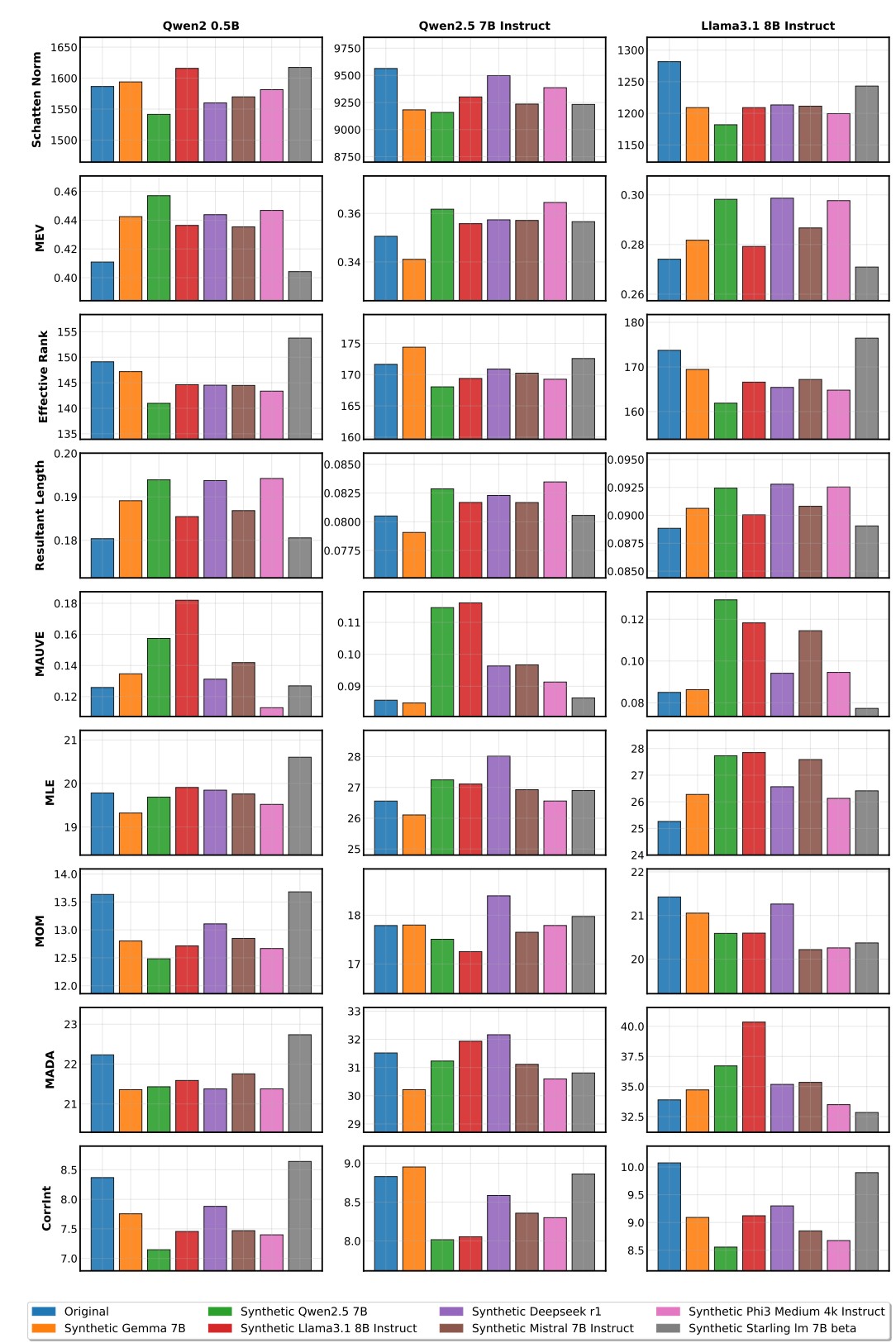

Figure 6: The ranking of eight generators models via tester models (Qwen2 0.5B, Qwen2.5 7B Instruct and LLama3.1 8B Instruct) and all geometric metrics.

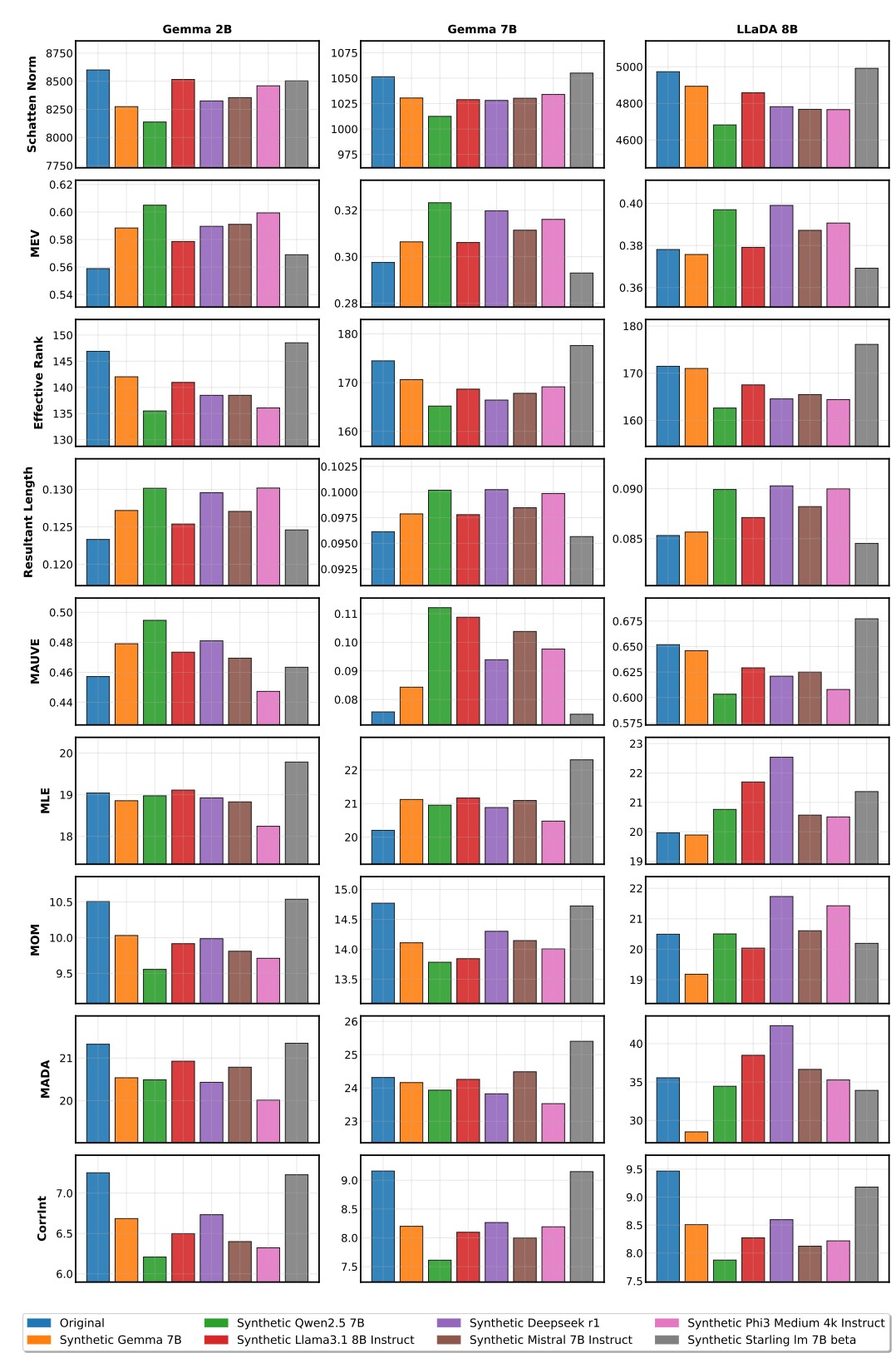

Figure 7: The ranking of eight generators models via tester models (Gemma 2B, Gemma 7B and LLaDA 8B) and all geometric metrics.

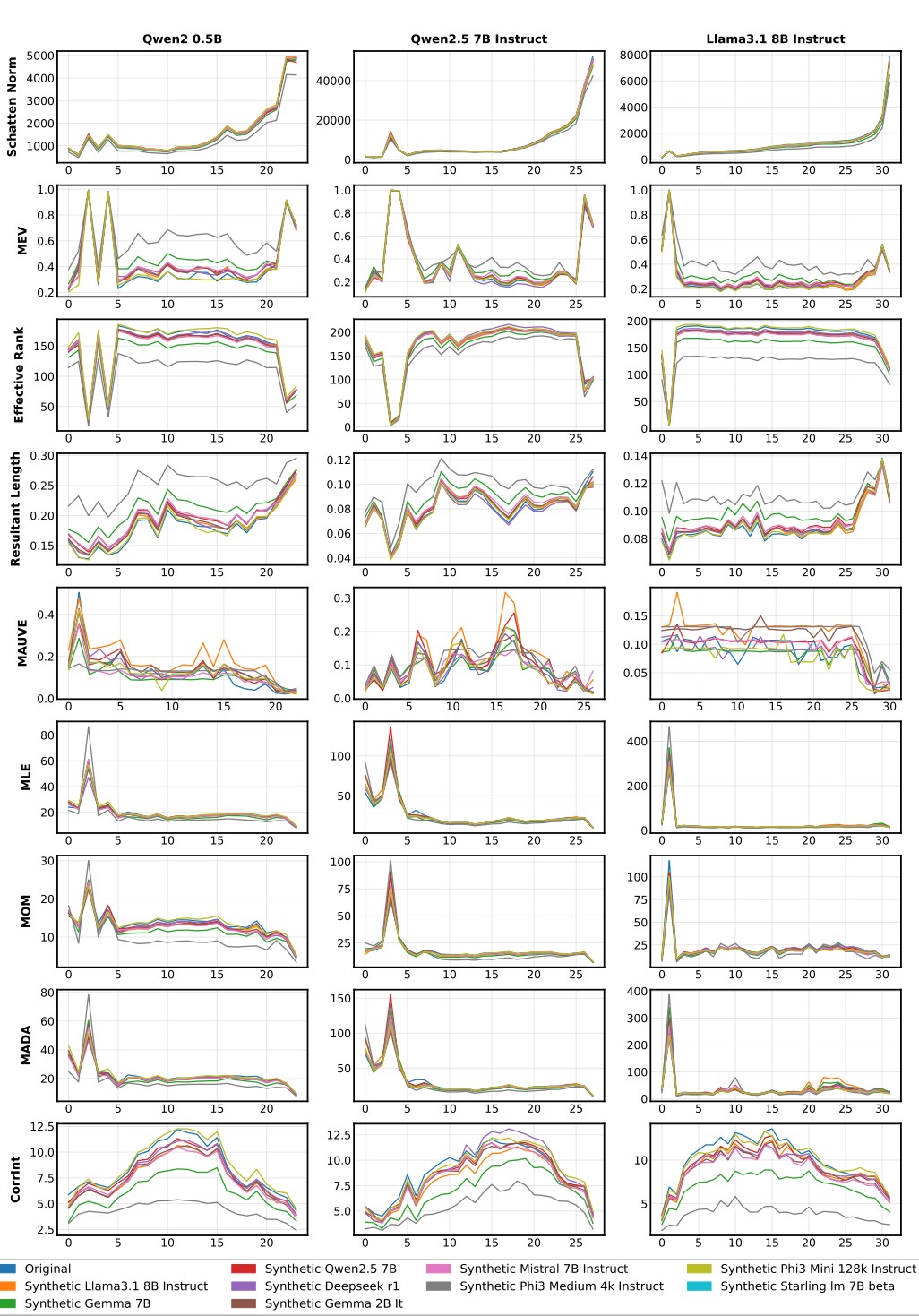

Figure 8: The layerwise ranking of ten generators models via tester models (Qwen2 0.5B, Qwen2.5 7B Instruct and LLama3.1 8B Instruct) and all geometric metrics.

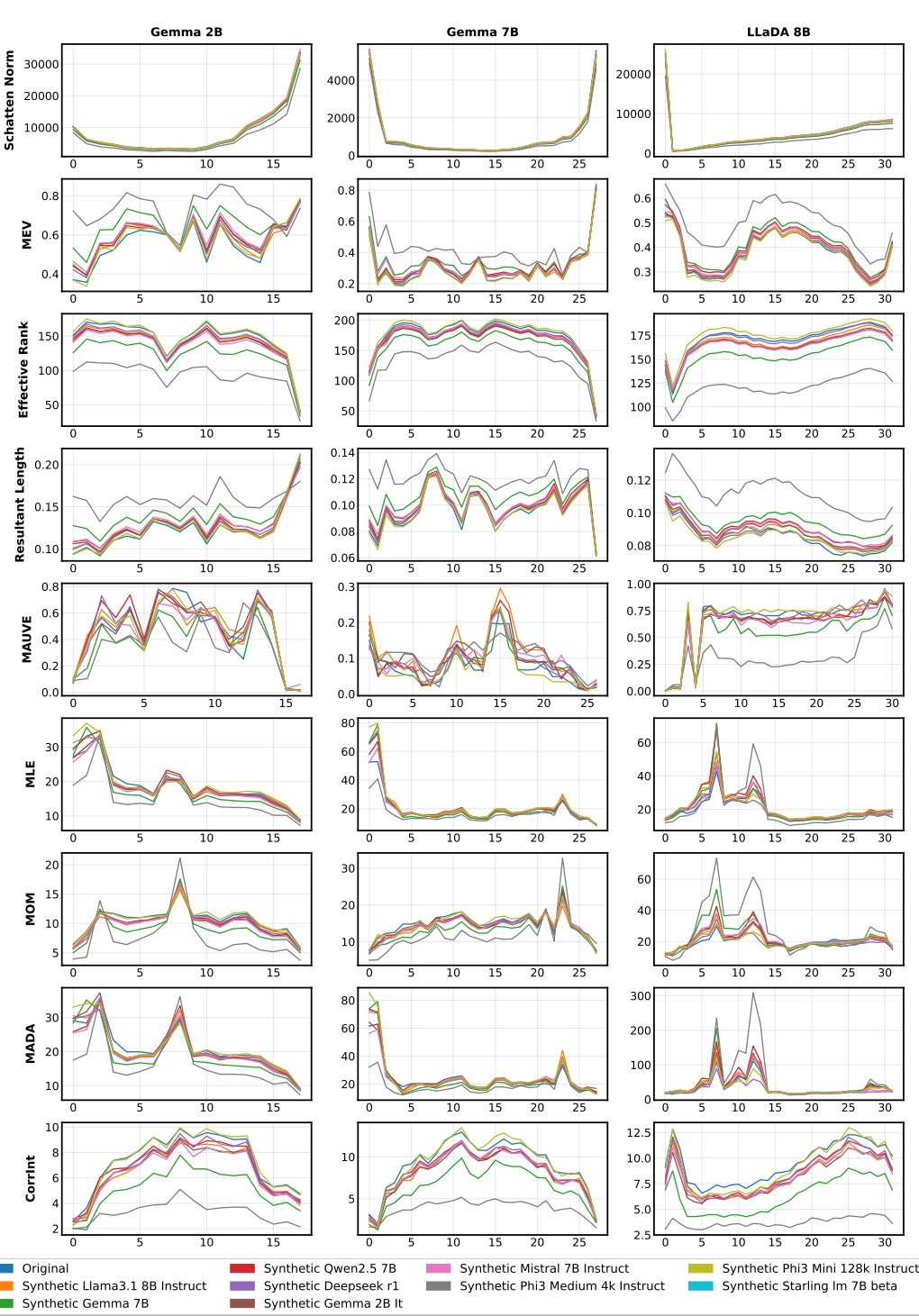

Figure 9: The layerwise ranking of ten generators models via tester models (Gemma 2B, Gemma 7B and LLaDA 8B) and all geometric metrics.

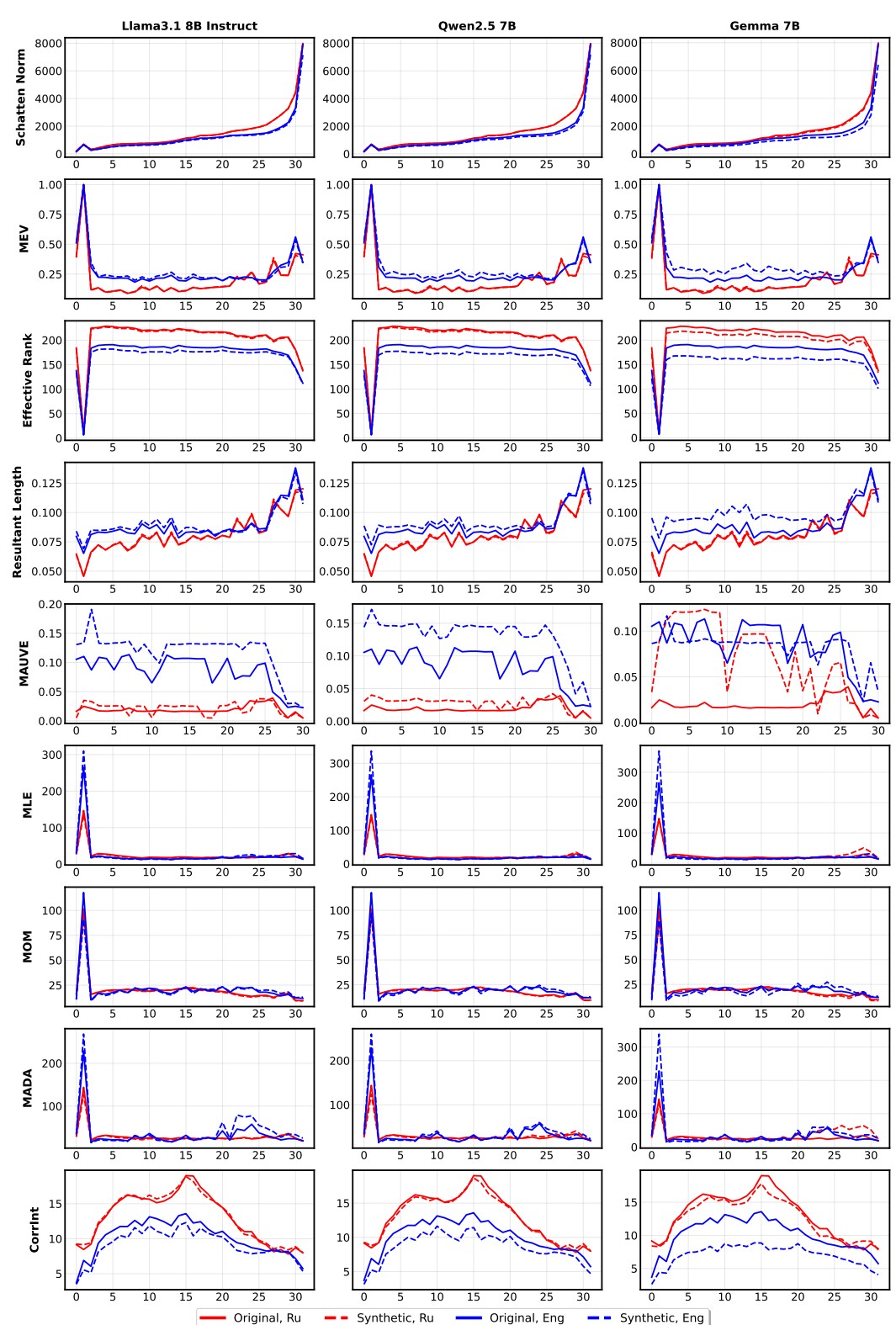

Figure 10: The comparison of Original and Synthetic Russian and English texts generated by Llama3.1 8B Instruct, Qwen2.5 7B Instruct and Gemma 2B and tested via Llama3.1 8B Instruct.

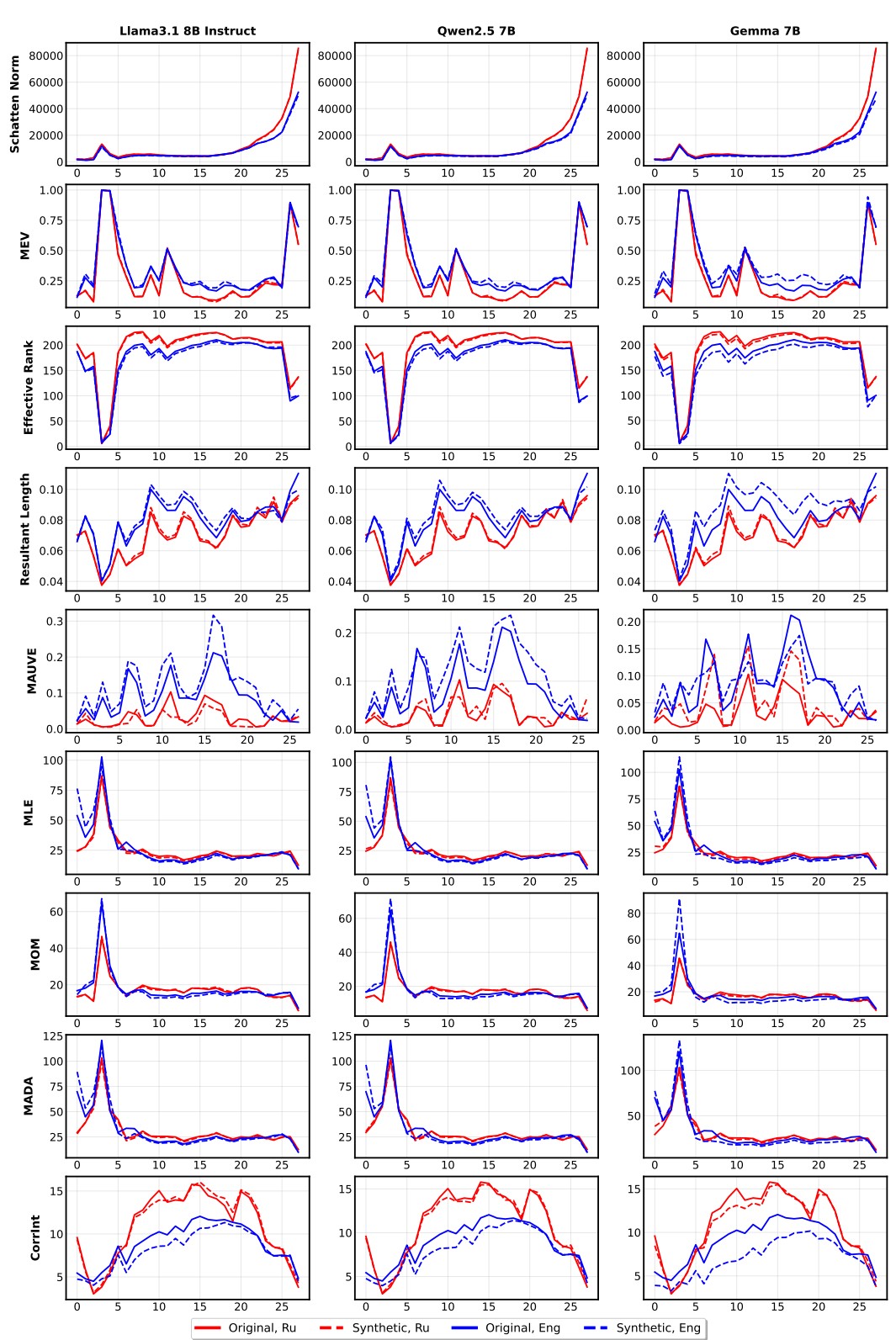

Figure 11: The comparison of Original and Synthetic Russian and English texts generated by Llama3.1 8B Instruct, Qwen2.5 7B Instruct and Gemma 2B and tested via Qwen2.5 7B Instruct.

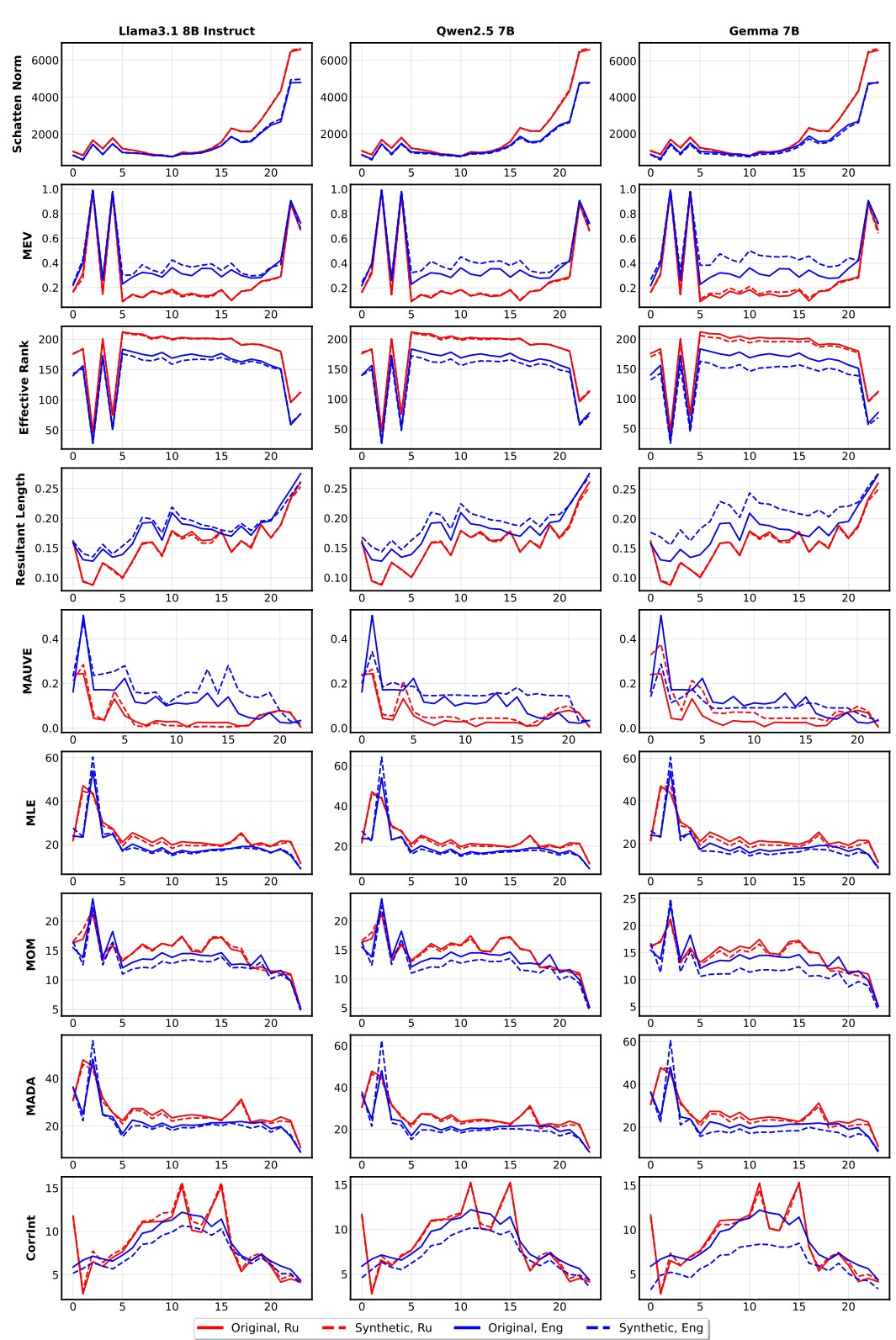

Figure 12: The comparison of Original and Synthetic Russian and English texts generated by Llama3.1 8B Instruct, Qwen2.5 7B Instruct and Gemma 2B and tested via Qwen2 0.5B.

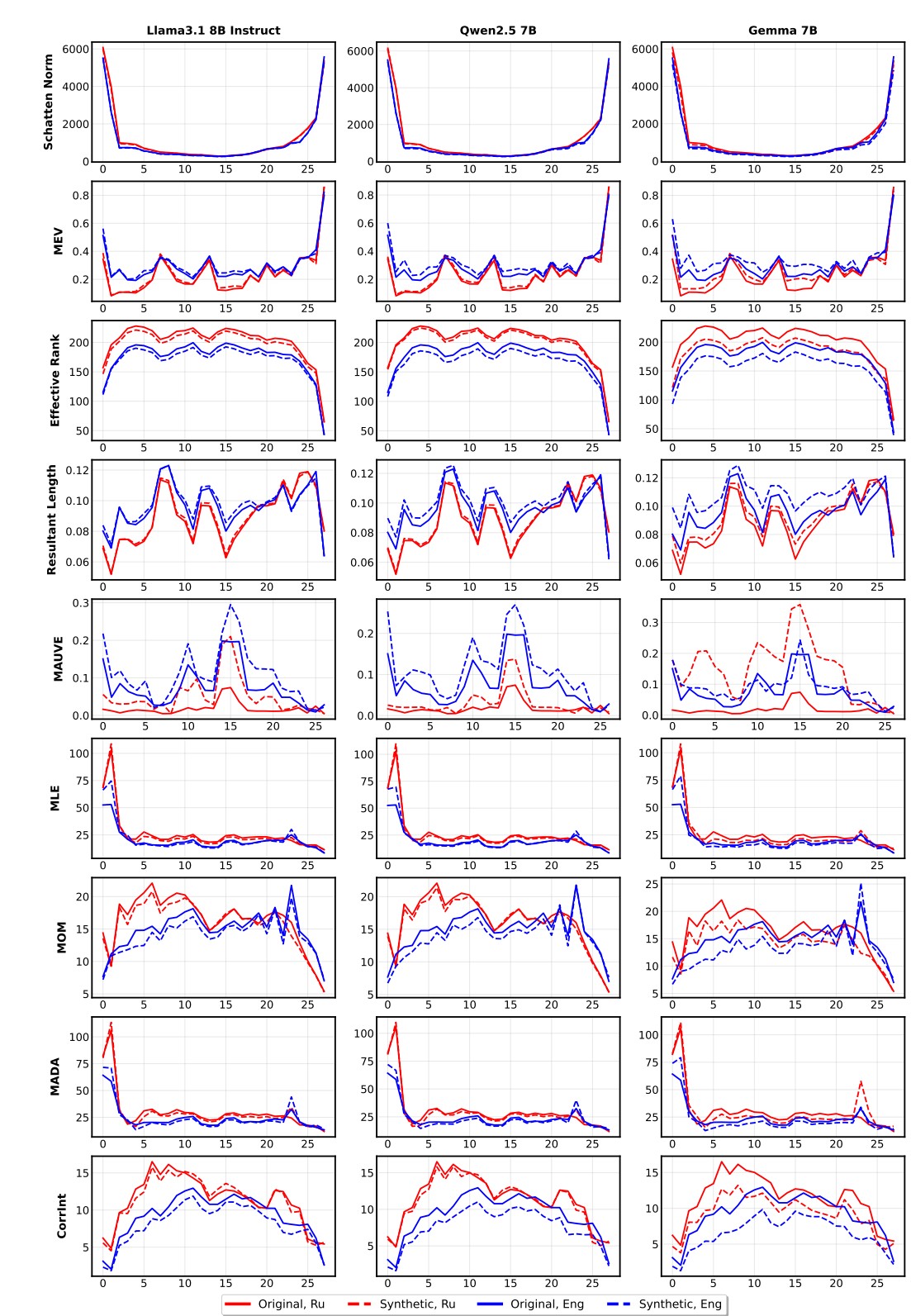

Figure 13: The comparison of Original and Synthetic Russian and English texts generated by Llama3.1 8B Instruct, Qwen2.5 7B Instruct and Gemma 2B and tested via Gemma 7B.

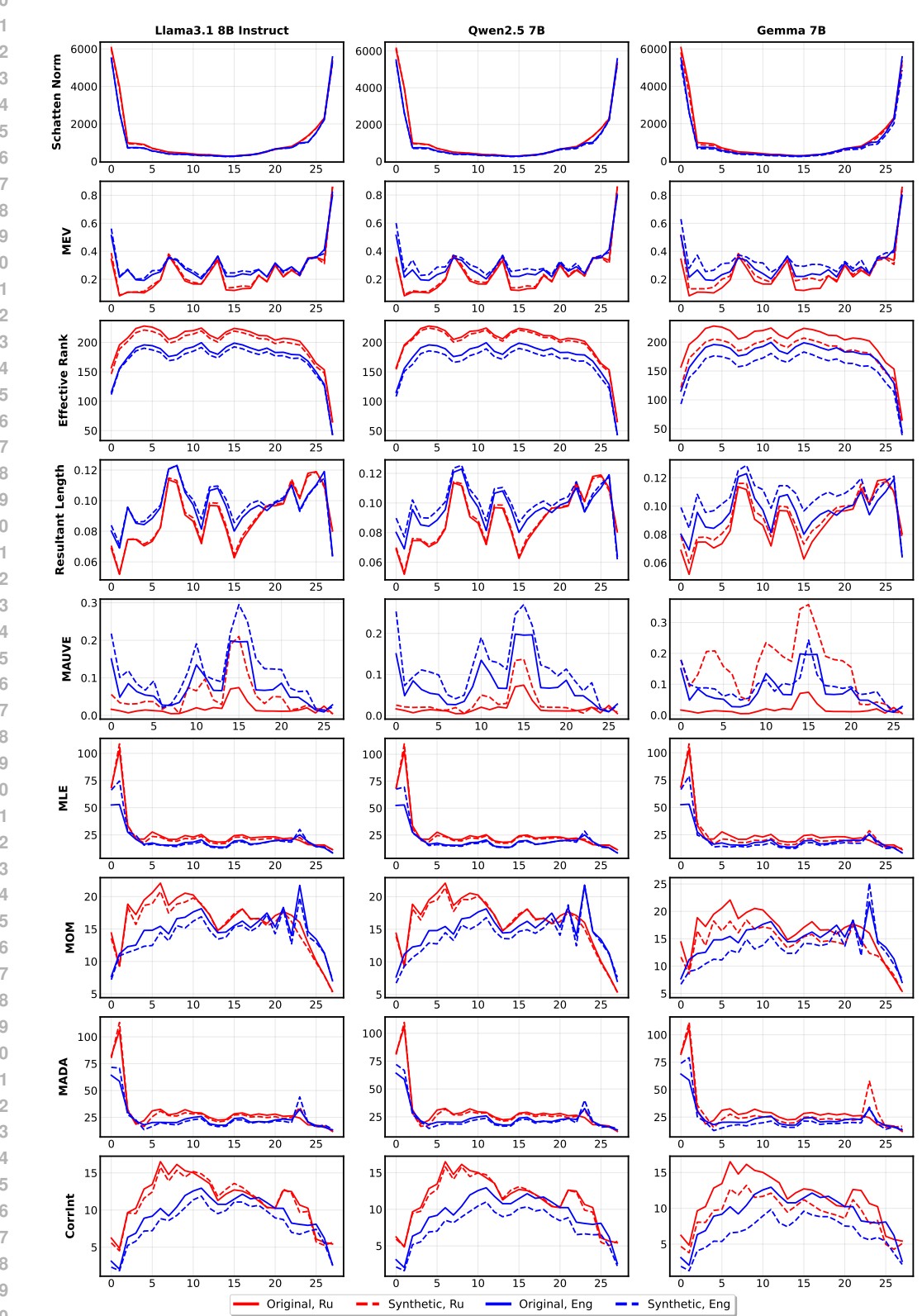

Figure 14: The comparison of Original and Synthetic Russian and English texts generated by Llama3.1 8B Instruct, Qwen2.5 7B Instruct and Gemma 2B and tested via Gemma 7B.

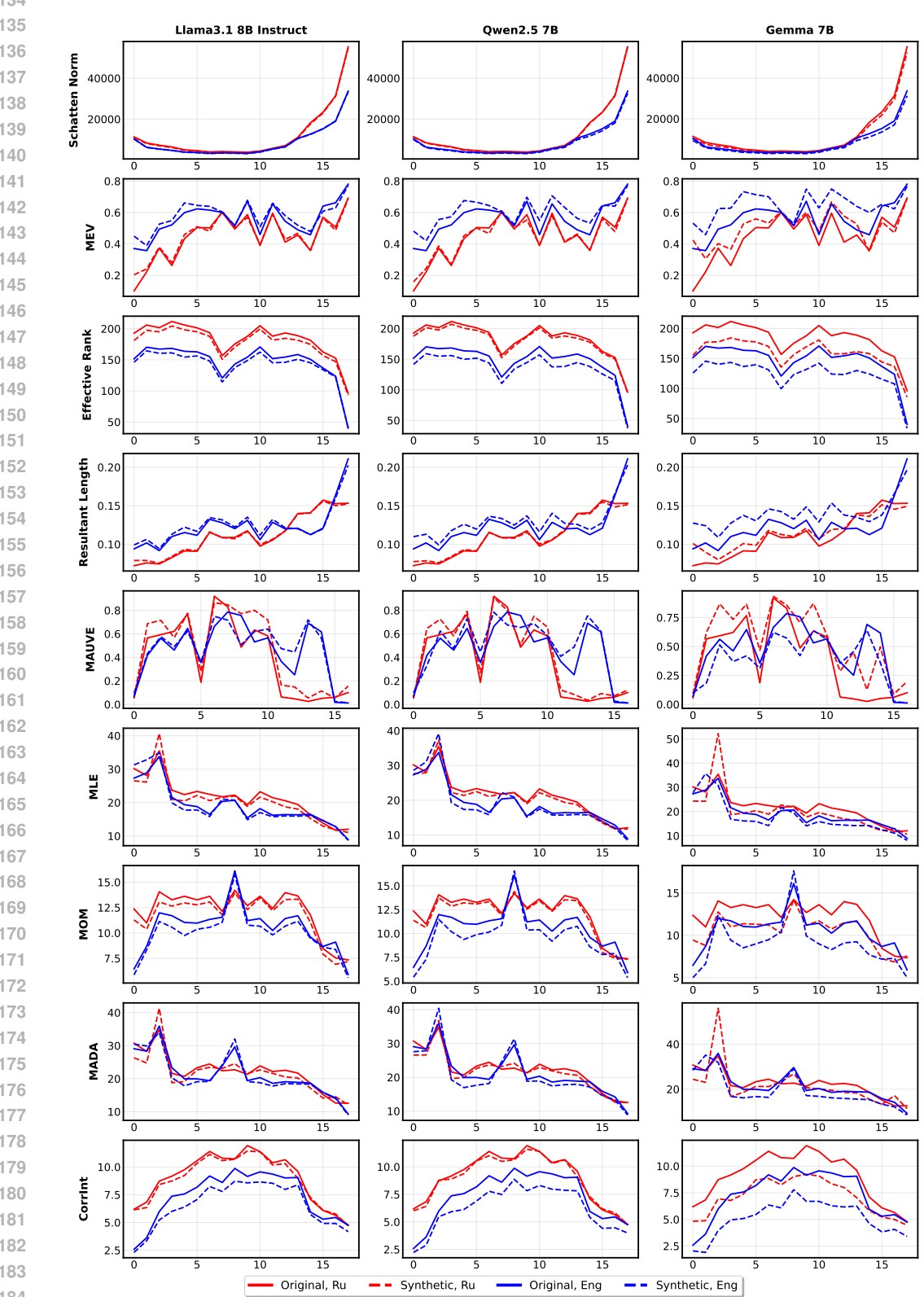

Figure 15: The comparison of Original and Synthetic Russian and English texts generated by Llama3.1 8B Instruct, Qwen2.5 7B Instruct and Gemma 2B and tested via Gemma 2B.

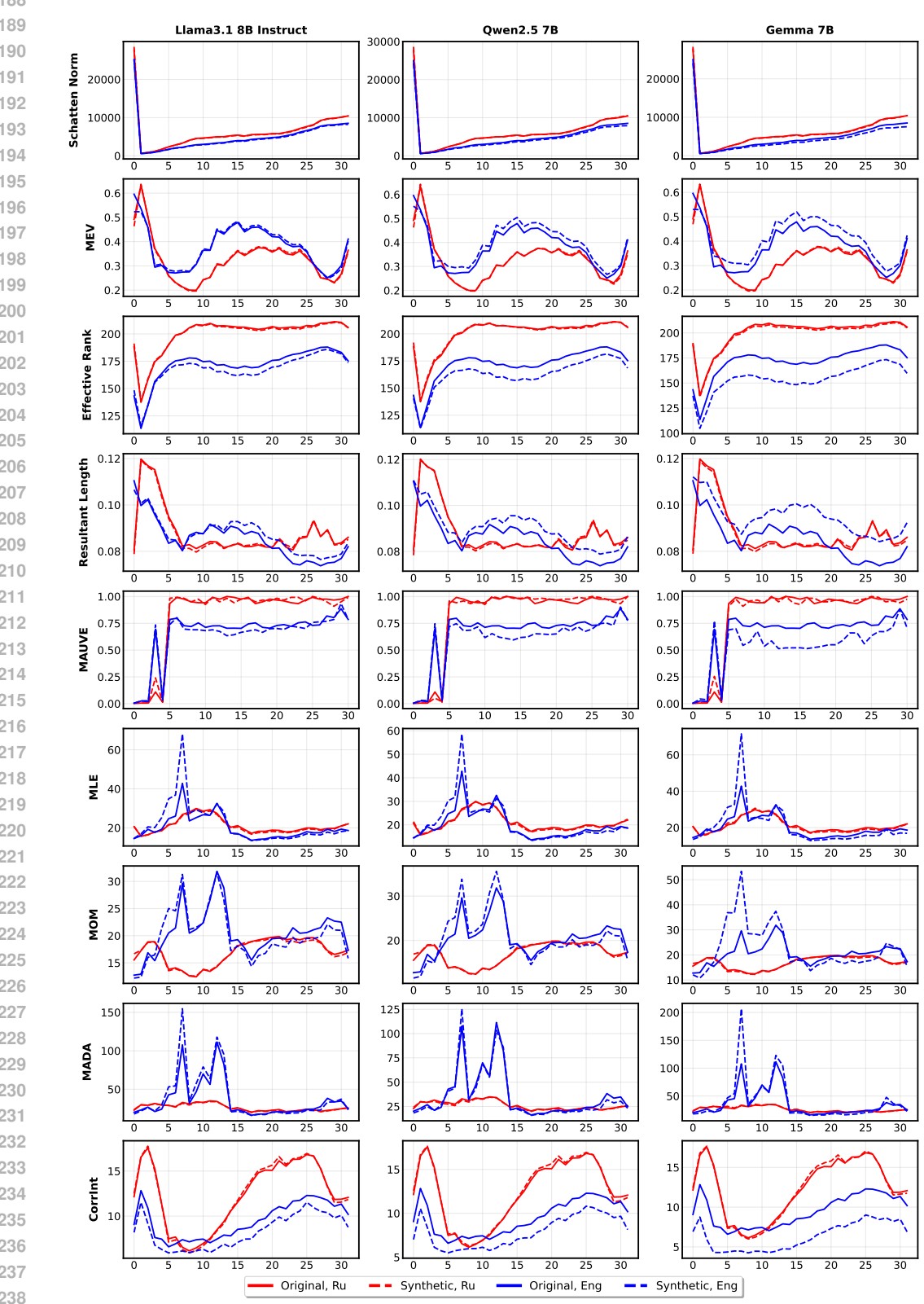

Figure 16: The comparison of Original and Synthetic Russian and English texts generated by Llama3.1 8B Instruct, Qwen2.5 7B Instruct and Gemma 2B and tested via LLaDA 8B.

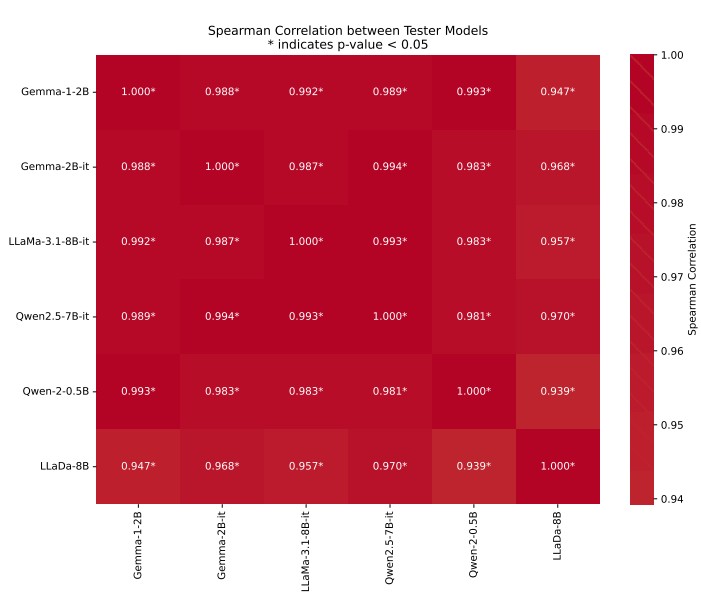

Figure 17: Spearman correlation among all $\mathcal{T}$ models for all geometric $\mathbf{R}$ scores.

