# OpenReview forum: "From Internal Representations to Text Quality: A Geometric Approach to LLM Evaluation"
_ICLR.cc/2026/Conference — Submitted to ICLR 2026_

### Official Review · Reviewer_ZTbS · 2025-10-25

**Soundness:** 3
**Presentation:** 3
**Contribution:** 3
**Rating:** 6
**Confidence:** 3

**Summary:**

This paper investigates the relationship between the geometric properties of internal representations within Large Language Models (LLMs) and the perceived quality (specifically, "naturalness") of the text they generate. The authors propose using a set of "tester" LLMs to analyze text produced by different "generator" LLMs. By extracting hidden states from the tester models processing the generated text, they calculate several geometric metrics: Schatten Norms, Maximum Explainable Variance (MEV), Effective Rank (ERank), Resultant Length, MAUVE (applied to internal representations), and various Intrinsic Dimensionality (ID) estimators (MLE, MOM, MADA, CorrInt).

**Strengths:**

The paper effectively connects the often separate fields of mechanistic interpretability (analyzing internal representations) and black-box evaluation (assessing output text quality). Showing that internal geometric properties reliably predict external quality is a significant conceptual contribution. The demonstration that diverse tester models (different sizes, architectures like autoregressive vs. diffusion) yield highly consistent rankings (Spearman > 0.947) for generator models based on internal metrics is a powerful result. This strongly supports the claim that these metrics capture inherent text properties.

**Weaknesses:**

The correlation analysis between the internal geometric scores and external text quality metrics (Figure 4) relies on scores aggregated across tester models, with the number of data points being the number of generator models (plus human text). As acknowledged by the authors, this limits the statistical power and prevents robust p-value calculation. Correlations based on individual text samples might provide stronger evidence.

**Questions:**

Regarding the correlation analysis in Figure 4: Would it be feasible to compute correlations at a finer granularity (e.g., per text sample, averaging metrics across layers but not across generators) to obtain more data points and increase statistical confidence in the relationship between internal geometry and external quality metrics?

---

> ### Author Response · Authors · 2025-11-22
>
> Thank you for your comment.
>
> Yes, we will perform per-sample evaluation on a new dataset to demonstrate applicability of our idea to other text domains.
>
> The setup will be following, we want to use titles and abstracts of scientific papers. Then we will generate new abstracts given titles. In this setup we will be able to explore another domain and another task apart from paraphrasing.
>
> We will include these results into camera ready version.
>
> For the current results we did not store per-example values, so, we will need to re-compute them again that is another reason to add one more dataset.

---

### Official Review · Reviewer_9fFU · 2025-11-01

**Soundness:** 2
**Presentation:** 2
**Contribution:** 2
**Rating:** 4
**Confidence:** 5

**Summary:**

The paper evaluates many metrics, including Schatten norms, Maximum Explainable Variance (MEV), Resultant Length, Effective Rank (ERank), Intrinsic Dimensionality (ID) variants, and MAUVE, based on text embedding from LLMs averaged across layers. The paper argues that ERank and ID are the most reliable “universal” indicators. Across six tester models (0.5B–8B), the authors report consistent rankings of generators’ text, suggesting these metrics capture properties of the text rather than idiosyncrasies of a particular tester. They also report correlations between internal geometry and external text metrics (e.g., BLEURT, GPT-PPL), and explore English, German, Russian movie-review paraphrases as a testbed.

**Strengths:**

1. Internal geometry of a small tester model can diagnose properties of text produced by other models. This enables reference-free evaluation, which is very important to supplement classic reference-based scores like ROUGE.
2. Rankings of generators are broadly consistent across heterogeneous tester models. This is actually very interesting observation I didn't expect.
3. The paper formalizes and compares many geometry metrics (MEV, Resultant Length, ERank, ID variants, Schatten norms) with precise definitions and layer-wise/averaged reporting.
4. The paper also presents comprehensive results of Spearman correlations between geometry and BLEURT/GPT-PPL/MAUVE.
5. It also extends to non-English text (German, Russian).

**Weaknesses:**

1. The evaluation is built around paraphrasing movie reviews to “synthetic” text with preserved meaning/length. This seems very narrow. Can you apply on real-world conversation prompt dialogue, such as UltraChat [1]? My concern is that the results in the paper are mainly due to dataset characteristics rather than text quality in general.
2. The results could cofound with each LLM text preprocess. In fact, if you generate responses with batch inference (like using vLLM), the padding could affect the resulting embedding. The individual embedding for a specific text would be different from that if using native inference one by one. How do you handle the artifacts in tokenization and padding for each LLM.
3. LLM-as-a-judge becomes a popular approach to judge generation quality. It is also found that it very correlates with human judgement [2]. How does any of the metrics evaluated in this paper correlate with LLM-as-a-judge. It would also investigate how the judge model would prefer its own generation, or the embedding is more model-agnostic metric?

References:
1. Ding et al. Enhancing Chat Language Models by Scaling High-quality Instructional Conversations
2. Chiang et al. Chatbot Arena: An Open Platform for Evaluating LLMs by Human Preference. ICML 24'

**Questions:**

1. How do the metrics studied in this paper correlate with one another? I think this investigation brings more value compared with PPL or AVG Len.
2. Also the different models yield identical rankings of texts generated by various generator. This seem contradictory with results self preference bias [1] using an LLM-as-a-judge. Is it because the embedding more model-agnostic to the input? Can you provide some explanations to reconcile the gap.
3. Recent studies [2, 3] find that generated text from different LLMs based on the same prompt are idiosyncratically different in word choices, semantics, and style. How do you see your findings on diversity metrics can be used as a tool to quantify the degree of difference in these cases. This can also be extended to detecting AI vs machine-generated text, or which LLM generates a specific text.

Reference:
1. Wataoka et al. Self-Preference Bias in LLM-as-a-Judge.
2. Sun et al. Idiosyncrasies in Large Language Models. ICML 25
3. Dunlap et al. Vibecheck: Discover and quantify qualitative differences in large language models. ICLR 25

---

> ### Author Response · Authors · 2025-11-21
> **Authors  Answer**
>
> **Thank you for your review. We also note that the full codebase will be released with the camera-ready version.**
>
> ---
>
> **W1.  The evaluation is built around paraphrasing movie reviews to “synthetic” text with preserved meaning/length**
>
> We share your concerns about the possible mixing of the dataset characteristics and text quality in general. We definitely will add one more dataset. We appreciate your offering of UltraChat; however, according to paper [1], this dataset consists of “instructional conversations”, LLM-generated dialogs, “which does not involve human queries”. Nevertheless, we appreciate and accept our suggestion of evaluating our metrics on the real-world texts, that is why we decided to choose  **DailyDialog - 13k**.
>
> ---
>
> **W2.  The results could cofound with each LLM text preprocess**
>
> Thank you for the comment. In our experiments, we consider each text separately, so there is no padding and we do not apply any other preprocessing for both synthetic and original texts,

---

> ### Author Response · Authors · 2025-11-21
> **W3. LLM-as-a-judge**
>
> We performed an LLM-as-a-judge experiment using Deepseek-v3 as our judge model. To compute win rates, we conducted pairwise comparisons of generated texts while switching text positions to mitigate positional bias. We employed three distinct prompts based on [5], resulting in 6k pairwise evaluations per prompt (18k evaluations total).
>
> Our initial observation reveals that the results are highly dependent on the prompt strategy, with the exception of mistral_7b_it and phi3-mini, which consistently demonstrated the lowest win rates across all prompts. Consequently, we averaged the win rates across all three prompts for each model.
>
> | Model                  | Column 1 (Score)       | Column 2 (Score)       | Column 3 (Score)       | Average |
> |------------------------|------------------------|------------------------|------------------------|---------|
> | llama31                | 68.4% (648/947)        | 79.3% (1731/2184)      | 85.9% (1608/1872)      | 77.9%   |
> | gemma-2-baku-2b-it     | 52.1% (495/950)        | 65.8% (1436/2184)      | 75.2% (1407/1872)      | 64.4%   |
> | original               | 93.5% (889/951)        | 65.6% (1433/2184)      | 47.5% (890/1873)       | 68.9%   |
> | qwen25                 | 53.5% (507/947)        | 64.5% (1408/2184)      | -                      | 59.0%   |
> | deepseek               | 59.6% (567/952)        | 49.6% (1084/2184)      | 50.2% (941/1873)       | 53.1%   |
> | starling-lm-7b-beta    | 32.8% (311/948)        | 38.1% (831/2184)       | 44.8% (839/1872)       | 38.6%   |
> | mistral_7b_it          | 31.7% (300/948)        | 26.4% (577/2184)       | 29.8% (557/1872)       | 29.3%   |
> | phi3-mini              | 8.2% (78/947)          | 10.8% (235/2182)       | 16.6% (311/1872)       | 11.9%   |
>
>
> Finally, we computed correlations between these average win rates and our metrics presented in Table 2. While we observe supporting evidence for ERank (which shows the strongest statistically significant correlation), the high variance and prompt dependence lead us to conclude that LLM-as-a-judge may not serve as a reliable reference metric.
>
> ### Correlations with Average Win Rate
>
> | Metric       | Spearman Correlation (p-value)     | Pearson Correlation (p-value)     |
> |--------------|------------------------------------|-----------------------------------|
> | Text_origin  | 0.444 (p = 0.232)                  | 0.396 (p = 0.292)                 |
> | Schatten     | 0.552 (p = 0.123)                  | 0.524 (p = 0.148)                 |
> | MEV          | 0.519 (p = 0.152)                  | 0.498 (p = 0.172)                 |
> | **ERank**        | -0.695 (p = 0.038)                 | -0.668 (p = 0.049)                |
> | Resultant    | -0.577 (p = 0.104)                 | -0.637 (p = 0.065)                |
> | MAUVE        | 0.318 (p = 0.404)                  | 0.394 (p = 0.293)                 |
> | MLE          | 0.025 (p = 0.949)                  | 0.083 (p = 0.831)                 |
> | MOM          | 0.176 (p = 0.651)                  | 0.177 (p = 0.649)                 |
> | MADA         | 0.519 (p = 0.152)                  | 0.532 (p = 0.140)                 |
> | CorrInt      | 0.261 (p = 0.498)                  | 0.242 (p = 0.531)                 |
>
> ### Ranked by Spearman Correlation (Absolute Value)
>
> | Metric       | Spearman Correlation               | Pearson Correlation               |
> |--------------|------------------------------------|-----------------------------------|
> | **ERank**        | -0.695*                            | -0.668*                           |
> | Resultant    | -0.577                             | -0.637                            |
> | Schatten     | 0.552                              | 0.524                             |
> | MEV          | 0.519                              | 0.498                             |
> | MADA         | 0.519                              | 0.532                             |
> | Text_origin  | 0.444                              | 0.396                             |
> | MAUVE        | 0.318                              | 0.394                             |
> | CorrInt      | 0.261                              | 0.242                             |
> | MOM          | 0.176                              | 0.177                             |
> | MLE          | 0.025                              | 0.083                             |

---

> ### Author Response · Authors · 2025-11-21
>
> **We attach prompt 1 used for LLM-as a Judge evaluation, which is based on [5]. For other prompts some criteria were added or removed without significant change in meaning.**
>
> ```System Prompt
> You are an impartial judge evaluating the naturalness of text.
>
> You will be given two texts, Text A and Text B.
>
> Your task is to determine which one is more natural, meaning it reads as if it was written by a native speaker.
>
> Follow these citeria to evaluate the text:
>
> Which text is more likely to be human written rather than generated?
>
> Does it look like that the text was written by a native speaker?
>
> Which text is more formal-natural?
>
> Is the word usage contextually appropriate and accurate and not overly complex?
>
> Is the vocabulary appropriate for the context and suitable for the topic of Movie reviews?
>
> Is the text free of repetitive or unnecessary content?
>
> In the text, are the sentences well connected and progressing naturally and logically?
>
> Did you use conjunctions appropriately to improve readability?
>
> Are the paragraphs in the text logically connected and progressing with each other?
>
> Is the entire article consistently progressing with the central theme as the focus?
>
> Does each paragraph of the article have a clear subtopic centered around the main idea?
>
> Is a consistent narrative tone and style maintained throughout the entire text?
>
> Is there no sudden change in tone in the context of the writing?
>
> Does the given text maintain a consistent style type (spoken language, written language, informal, formal, etc.)?
>
> Does author consistently use abbreviations and acronyms when necessary?
>
> Is it a clear text that does not excessively use uncommon or complex words?
>
> Are unfamiliar and difficult words used appropriately in context?
>
> Is the given text clearly structured without excessively complex sentence structures?
>
> Ignore factual accuracy focus on the naturalness of the language.
>
> Is the text coherent?
>
> You MUST output your judgment in a strict JSON format:
>
> {
> "winner": "A" or "B",
>
> "overall_A": how natural and human-like does this text feel on a scale from 1 to 5?,
>
> "overall_B": how natural and human-like does this text feel on a scale from 1 to 5?,
>
> "reason": "A brief, 1-2 sentence explanation for your choice."
> }```

---

> ### Author Response · Authors · 2025-11-21
>
> **Q1.  How do the metrics studied in this paper correlate with one another? I think this investigation brings more value compared with PPL or AVG Len.**
>
> Thank you for the question! Actually, we have the correlation analysis of geometric metrics on Figure 3 and the correlation analysis of the geometric metrics with text quality metrics on Figure 4. The results descriptions are provided in Sections 4.2 (lines 337-344) and Section 4.4 (lines 440-452), respectively.
>
> ---
>
> **Q2.   Also the different models yield identical rankings of texts generated by various generator. This seem contradictory with results self preference bias [1] using an LLM-as-a-judge. Is it because the embedding more model-agnostic to the input? Can you provide some explanations to reconcile the gap.**
>
> Thank you for the question! This paper [2] presents that some models have noticeable self-preference bias (as in Figure 1, GPT-4, Vicuna-13b, Koala-13b), but some models do not have (as in the same Figure 1, other models have very small self-preference bias). Moreover, the authors introduce the term “human-preffered” answer, but do not provide the criteria for its determination. All these make it difficult to conduct the analogical experiment; The authors also do not mention it, but the factological precision in answers may also play a role during evaluation, it is when a model relies on facts from the training data and prefers those facts on which it was trained. So, if a model finds more facts in its response than in another model’s generated response, it may choose itself only based on this fact.
>
> ---
>
> **Q3.  Recent studies [2, 3] find that generated text from different LLMs based on the same prompt are idiosyncratically different in word choices, semantics, and style. How do you see your findings on diversity metrics can be used as a tool to quantify the degree of difference in these cases. This can also be extended to detecting AI vs machine-generated text, or which LLM generates a specific text.**
>
> Thank you for the question and providing interesting papers!
>
> Paper [3] presents the exploration of models’ unique patterns in their outputs, referred as idiosyncrasies in LLMs, while we propose the geometrical metrics for ranging models. The authors of [3] address the common issue of LLM-as-a-judge, which we also will add to our paper: each LLM considers its own output as better than the outputs of other LLMs. We conduct similar experiment and our results show the following insight: 1) some models prefer their responses more frequently than the outputs of other models; 2) the ranging of models in this experiment highly depends on the prompt. However, it would be interesting to investigate idiosyncrasies in more detail in the context of their potential influence on our metrics, so, we will conduct an additional experiment and add it to the final submission version of our paper.
>
> To some extent, similar experiments are presented in the article [4] - the authors investigate “vibes” of Llama, GPT, Gemini and quantitatively measure the utility of each vibe. However, the authors highlight in limitations multiple times can lead to different vibes and different results, making it harder to reproduce findings exactly. However, we will add a similar experiment to our paper by final submission.
>
>
> **References:**
>
> [1] Ding, Ning, et al. "Enhancing chat language models by scaling high-quality instructional conversations." Proceedings of the 2023 Conference on Empirical Methods in Natural Language Processing. 2023.
>
> [2] Wataoka, Koki, Tsubasa Takahashi, and Ryokan Ri. "Self-preference bias in llm-as-a-judge." arXiv preprint arXiv:2410.21819 (2024).
>
> [3] Sun, Mingjie, et al. "Idiosyncrasies in large language models." arXiv preprint arXiv:2502.12150 (2025).
>
> [4] Dunlap, Lisa, et al. "Vibecheck: Discover and quantify qualitative differences in large language models." arXiv preprint arXiv:2410.12851 (2024).
>
>
> [5] Navigating the Path of Writing: Outline-guided Text Generation with Large Language Models
>
> ---
>
>
> **We sincerely thank the reviewer for their thoughtful comments and hope we have addressed the concerns raised. If the reviewer is satisfied with our responses and finds the work has strong merit for publication, we would greatly appreciate a score reconsideration.**

---

### Official Review · Reviewer_at29 · 2025-11-01

**Soundness:** 2
**Presentation:** 2
**Contribution:** 2
**Rating:** 2
**Confidence:** 4

**Summary:**

This paper proposes to evaluate the quality of text generated by large language models (LLMs) through geometric analysis of internal model representations.Specifically, the authors compute several geometric metrics—Maximum Explainable Variance (MEV), Effective Rank (ERank), Intrinsic Dimension (ID), Schatten Norm, and MAUVE—from the hidden states of multiple “tester” LLMs while processing text generated by eight “generator” LLMs. The main observation is that different tester models (ranging from 0.5B to 8B parameters) produce nearly identical rankings of generator quality based on these geometric metrics. The authors further report that some of these metrics (notably ERank and ID) correlate with conventional text quality measures such as BLEURT and GPT Perplexity, claiming that these geometric features can serve as reference-free proxies for text naturalness.

**Strengths:**

1.The goal of establishing reference-free automatic text evaluation is important and timely, given the rapid scaling of LLMs and the impracticality of human annotation.
2.The idea of connecting internal geometry of representations with external text quality is conceptually interesting
3. The study consistently compares multiple generators and testers across languages, and provides explicit mathematical definitions for all geometric metrics.

**Weaknesses:**

1. The work primarily aggregates existing geometric and evaluation metrics (ERank, ID, MAUVE, MEV) and computes correlations.It reads more like an experimental survey report than a scientific contribution.The paper lacks a coherent theoretical foundation or an explanatory mechanism linking geometry to text quality.
2. The key claim—that different tester models produce similar rankings of generator quality—is unsurprising and of modest insight.Such ranking consistency could trivially arise from shared lexical or fluency statistics rather than any intrinsic geometric property.
3. The study does not verify whether the metrics can capture fine-grained differences in text quality, thus the proposed framework lacks predictive utility. And no variance estimation, confidence intervals, or sensitivity analysis are reported.
4. The Introduction section is logically confused, mixing philosophical remarks (“beauty vs. utility”) with technical motivations. It fails to establish a clear problem statement, research question, or hypothesis.
5. The authors claim that Intrinsic Dimension and Effective Rank are “universal” quality measures, yet this is based on small-scale correlations from limited data.Non-English experiments even contradict their claim (smaller human–synthetic gap), but this inconsistency is ignored rather than analyzed.

**Questions:**

1.	Why were relatively small open-source models (e.g., Starling, Phi, Mistral) selected instead of stronger or more diverse systems like GPT-4, Claude, or Gemini?
2.	How do you ensure that the observed ranking consistency across tester models is not driven by superficial lexical or stylistic cues (e.g., token frequency, sentence length) rather than intrinsic geometric structure?
3.	Did you analyze which layers are most correlated with text quality instead of simply averaging across all layers? Could layer-specific behavior be more informative?
4.	Have you tested whether the geometric metrics are stable across random seeds, prompts, or domains?

---

> ### Author Response · Authors · 2025-11-22
>
> **Thank you for your review. We also note that the full codebase will be released with the camera-ready version.**
>
> ---
>
> **W1: The work primarily aggregates existing geometric and evaluation metrics (ERank, ID, MAUVE, MEV) and computes correlations.It reads more like an experimental survey report than a scientific contribution. The paper lacks a coherent theoretical foundation or an explanatory mechanism linking geometry to text quality.**
>
> We understand your concerns regarding the lack of links between geometry and text quality. However, we can not entirely share them according to the following reasons:
> We focus on the ranking property that various models rank different texts in the same order. This property brings more insights than pure knowledge of the connection of geometric characteristics and text quality, given the foundation of further research on LLMs’ text representations.
>
> Nevertheless, we have established the presence of linear and monotonic dependence of text quality metrics on geometry metrics, for example, Figure 4 shows strong direct relationship between Resultant Length and MEV with GPT PPL and inverse relationship with Effective Rank.
>
> ---
>
> **W2. The key claim—that different tester models produce similar rankings of generator quality—is unsurprising and of modest insight. Such ranking consistency could trivially arise from shared lexical or fluency statistics rather than any intrinsic geometric property.**
>
> Thank you for your comment! We will highly appreciate it if you provide papers supporting the fact that different tester models produce similar rankings of generator quality as has been reported in papers. Otherwise your accusation of unsurprising insight seems a bit subjective and unfounded.
>
> Our experiment with the correlation calculation between geometric metrics and text quality metrics show that our geometric metrics capture more than only lexical or fluency statistics. Otherwise the correlation between them and GPT-PPL (reflecting fluency) and ROUGE (reflecting lexical similarity) will be stronger (closer to 1) than shown in Figure 4.
>
> ---
>
> **W3. The study does not verify whether the metrics can capture fine-grained differences in text quality, thus the proposed framework lacks predictive utility. And no variance estimation, confidence intervals, or sensitivity analysis are reported.**
>
> Thank you for the essential note! We are working on the addition of one more dataset and will add the results to the final version of the paper.
>
> ---
> **W4. The Introduction section is logically confused, mixing philosophical remarks (“beauty vs. utility”) with technical motivations. It fails to establish a clear problem statement, research question, or hypothesis.**
>
> Thank you for the comment. We treated them as synonyms in the context of the text. However, if you found their usage in our paper confusing, we will rewrite the whole section to the camera-ready paper version.
>
> ---
> **W5.  The authors claim that Intrinsic Dimension and Effective Rank are “universal” quality measures, yet this is based on small-scale correlations from limited data. Non-English experiments even contradict their claim (smaller human–synthetic gap), but this inconsistency is ignored rather than analyzed. [Eng, Ru, De]**
>
> Thank you for the essential note! We agree that the proposed statement of universality is too strong and we will edit this part. Nevertheless, as we have already mentioned, we will add experiments for one more dataset, which will help us to straighten our research. We also would like to examine the cases with different languages in more detail; however, the main part of data in the world is in English, complicating fair comparison of the results obtained for English datasets with the datasets on other languages.

---

> ### Author Response · Authors · 2025-11-22
>
> **Q1 Why were relatively small open-source models (e.g., Starling, Phi, Mistral) selected instead of stronger or more diverse systems like GPT-4, Claude, or Gemini?**
>
>
> Thank you for the question. We chose the models that are available for analysis of their representations. Commercial APIs such as GPT-4, Gemini or Claude are not open-source models thus we are unable to analyze internal hidden representations on which our work is based. Moreover, models of such a huge scale as GPT-4, Claude, Gemini, require cluster-level computational resources that are beyond the means of most academic research environments.
>
> ---
>
> **Q2 How do you ensure that the observed ranking consistency across tester models is not driven by superficial lexical or stylistic cues (e.g., token frequency, sentence length) rather than intrinsic geometric structure?**
>
> Thank you for the question. Our experiment with the correlation calculation between geometric metrics and text quality metrics show that our geometric metrics capture more than only lexical or fluency statistics, because the correlation between them and GPT-PPL (reflecting fluency) and ROUGE (reflecting lexical similarity) are not very close to 1, which is shown in Figure 4.
>
> ---
> **Q3 Did you analyze which layers are most correlated with text quality instead of simply averaging across all layers? Could layer-specific behavior be more informative?**
>
> We thank you for your comment, layer-specific behavior could be more informative. This can be verified by performing binary classification: whether the text is generated or not, and seeing which metric values predict better – averaged or on a specific layer, but it is a task of artificial text detection, and that is not our focus.
>
> ---
>
> **Q4 Have you tested whether the geometric metrics are stable across random seeds, prompts, or domains?**
>
> Thank you for your question! These keys are important for stable performance, meaning that small changes in prompts cause small changes in texts and in models’ representations. However, this stability contradicts with your W3, because there you insist that with this property our proposed framework lacks predictive utility. We kindly ask you to provide more detailed explanations on thispair of question and weakness as their discussion will help us to improve our paper.

---

### Official Review · Reviewer_wULi · 2025-11-01

**Soundness:** 2
**Presentation:** 2
**Contribution:** 2
**Rating:** 4
**Confidence:** 4

**Summary:**

Previous approaches using LLMs (referred to as test model $T$ in this paper) to evaluate the quality of text generated by other LLMs (referred to as generation model $G$ in this paper) typically require task-specific and scenario-specific customized annotation guidelines and metrics. Some methods also necessitate annotating reference datasets for training the evaluation model $T$ or providing reference responses.

This paper argues that such tedious customization for each individual task is suboptimal. Therefore, the authors propose leveraging the internal states of model $T$, using geometric characteristics of these internal representations as the basis for ranking the quality of external text generated by $G$. The paper primarily employs metrics including Maximum Explainable Variance, Effective Rank, Intrinsic Dimensionality, MAUVE score, and Schatten Norms.

Experimental results demonstrate that ranking text generated by $G$ based on the internal states of $T$ produces consistent text quality orderings across different $T$ models, indicating the reliability of the proposed internal state metrics for measuring external text quality.

**Strengths:**

The proposed approach of utilizing the internal states of model $T$ to evaluate the quality of text generated by $G$ is well-motivated and promising. It can effectively alleviate the burden of task-specific customization. However, the paper lacks discussion on a comprehensive range of text quality dimensions, which limits its broader applicability.

**Weaknesses:**

1. Insufficient definition of text quality: The paper lacks a rigorous and formal definition of text quality. Relying solely on "naturalness" as the text quality indicator is inadequate. Given that the authors criticize task-specific evaluation criteria as suboptimal, they should provide a method that is universally applicable across different tasks with effective generalization capabilities. However, the paper fails to clearly demonstrate through experiments which specific aspects of text quality each metric captures.

2. Limited task scope: Line 161 mentions using model $G$ for text paraphrasing. This raises the question of whether the proposed method is merely an evaluation metric for paraphrase tasks and cannot generalize to other text generation tasks.

3. Lack of baseline comparisons: The paper does not compare the proposed metrics with existing evaluation methods under the same experimental settings as shown in Figures 1 and 2.

4. Missing critical details:
    1. What decoding parameters (e.g., temperature, top-p, top-k) are used when model $G$ performs paraphrasing?
    2. What is the distribution of text lengths in the evaluation? Is the method effective for longer texts? This is particularly important given that current LLMs routinely generate thousands or even tens of thousands of tokens.

**Questions:**

1. The paper employs multiple metrics for evaluating internal representations. Could the authors clarify: (a) What are the motivations and theoretical foundations for selecting these specific metrics? (b) What distinct aspects of text quality does each metric capture? (c) Are these metrics collectively comprehensive in covering different dimensions of text quality, or are there potential gaps?

---

> ### Author Response · Authors · 2025-11-22
>
> Thank you for the detailed review and constructive feedback on our paper.
>
> **W1. Insufficient definition of text quality**
>
> Thank you for the essential note! We admit that relying only on “naturalness” seems doubtful. Nevertheless, we cannot agree that it is possible to introduce a universal term of “text quality” across diverse domains. We suppose that the second criteria after naturalness can be factological correctness and logical coherence. However, we can not provide universal definitions that will be suitable for different domains.
>
> Moreover, there are similar perceptual metrics which measure generated data quality such Frechet distance for Images, PESQ for audio etc. For text such metrics usually require references BERTScore, ROUGE. One of our contributions is that we propose metrics that correlate with BERTScore and ROUGE and do not require reference.
>
> In our paper, we use the empirical definition text quality (recognized more natural), which does not limit us, as the text quality can be quantitatively estimated by human labeling or through perplexity evaluation.
>
> Providing the comprehensive, formal and rigorous definition of the intuitive concepts of “Text quality” and “text naturalness” or the ultimate list of text quality dimensions is a difficult problem at the intersection of language processing, linguistics and philosophy, and is out of scope of our research.
>
> We refer to [1-3] for the thorough discussion of the formal definition of “text quality”.
>
> As well, it was not the goal of our paper to:
> * define the ultimate list of aspects, distinguishing texts of good and bad quality was not the goal of our research.
> * explicitly interconnect each metric to list of some text quality properties.
>
> However, we outlined the connection between the metrics, presented in the text, and text diversity:
>
> * Good quality of text correlates with the text diversity (An extensive vocabulary and complex grammar structures), as more diverse texts may express different aspects, details and nuances of the presented information.
>
> * Text diversity strongly correlates with contextual embeddings diversity as the semantic closeness of tokens is closely related to the closeness of its contextual representations. Thus, the diversity of contextual representations measures the text diversity.
>
> **References:**
>
> [1] Hassan, Syed Zohaib, Pierre Lison, and Pål Halvorsen. "Enhancing Naturalness in LLM-Generated Utterances through Disfluency Insertion." arXiv preprint arXiv:2412.12710 (2024).
> [2] Xia, Sirui, et al. "Ground every sentence: Improving retrieval-augmented llms with interleaved reference-claim generation." Findings of the Association for Computational Linguistics: NAACL 2025. 2025.
> [3] Jaydip Sen, Saptarshi Sengupta, and Subhasis Dasgupta. Advancing decoding strategies: Enhancements in locally typical sampling for llms. arXiv preprint arXiv:2506.05387, 2025.
>
> ---
>
> **W2 Limited task scope**
> Thank you for the essential note! We will conduct an additional experiment with words’ perturbation and will add it to the paper by the camera-ready submission.
>
> ---
>
> **W3 Lack of baseline comparisons**
>
> Thank you for your comment. We will appreciate your recommendation of the baseline geometric measure based intermediate representations to assess generated text quality, because our research does not provide other metrics in addition to the considered. While other geometric measures exist, we focused on those, which have interpretable meaning in terms of text quality as explained in Table 2. As the text quality baseline metrics we use ROUGEL BLUERT GPT PPL.
>
> Nevertheless, our main contribution is the empirical analysis of reference-free text quality metrics, their intercorrelation and the correlation between them and text quality. So, we do not focus on developing a particular method for the estimation of text quality. We state that the metrics, proposed in the article, can be informative features of the text quality.
>
> ---
>
> **W4 Missing critical details:**
>
> 1) Decoding parameters are important only for text generation, which was used for parallel  texts creation. For each model we used greedy decoding with temperature set to 0.1. We used ollama framework for text generation.
>
>  2) Length for generated sentences is 255.
>
> ---

---

> ### Author Response · Authors · 2025-11-22
>
> **Q1 (a)**
> Let us start with the **motivation** of our research. We were inspired by papers:
> * Stable Anisotropic Regularization, https://arxiv.org/pdf/2305.19358v3
> * Intrinsic Dimension Estimation for Robust Detection of AI-Generated Texts, https://proceedings.neurips.cc/paper_files/paper/2023/file/7baa48bc166aa2013d78cbdc15010530-Paper-Conference.pdf
>
> All the proposed metrics, except Schatten-p norms, are united by the same idea:
> * Good quality of text correlates with the text diversity (an extensive vocabulary and complex grammar structures), as more diverse texts may express different aspects, details and nuances of the presented information.
> * Text diversity strongly correlates with contextual embeddings diversity as the semantic closeness of tokens is closely related to the closeness of its contextual representations. Thus, the diversity of contextual representations measures the text diversity.
> * Additionally we developed a formal model, which justifies usage of effective rank and MEV as text quality metrics.
>
> Turning to **theoretical foundation**: let us introduce two core suppositions for our explanation:
> 1) Text of good quality is **brief**: it means that the same information, presented in the text, can not be expressed with the smaller number of tokens according to the language rules.
> Let us denote with ILen(T) the length of the brief text T’, containing the same information as the text T.
> 2) All the semantic operations with the tokens can be represented as the linear operations with its contextualized embeddings.
> This supposition makes sense, as it was noticed that linear operations with token embeddings tend to correspond to semantic operations: e.g. famous: king embedding - man embedding + woman embedding is very close to queen embedding.
>
> **Proposition:** rank of the embedding matrix, obtained from text T does not exceed the length ILen(T).
>
> **Proof:** indeed, let us consider the brief text T’ of ILen(T), containing the same information as T.
> Then, according to the supposition 2, the fact that information from each token of text T can be obtained from T’, can be expressed that contextualized embedding of each token of text T is a linear combination of
>
> Thus, the embedding matrix of text T has rank no more than len(T’) = ILen(T)
>
> **Proposition meaning:**
> 1) It provides an idea that zero linear combinations of token embeddings are the sign of the bad text quality. Here we list some text defects, which will result in zero linear combinations of embeddings:
> * **Contradictions:** “He is a good person and he is a bad person”, contextualized embeddings of antonyms, referring to the same object, have 0 sum, indicating logical contradiction in the text
>
> * **Information repeat:** “The cat is red. The cat is red”, as the word “red” refers to the same object “the cat”, contextualized embeddings of the words ‘red’ would be close to similar, thus, its difference would be close to zero
>
> 2) Full ranked embedding-matrices refer to the effective expression of text information.
> Consequently, if text is brief (in the sense mentioned above) and has a lot of tokens (due to it contains a lot of information, it is **diverse**), then the rank of the embedding matrix would be high.
>
> However, we can not use the raw rank as it’s not continuous and numerically stable metric, instead, we refer to numerically stable rank bounds:
> Effective rank as the lower bound on the rank and MEV as the upper bound on the rank.
>
> ---
>
> **Q1 (b)** Thank you for the question. The correspondence between each metric and natural text property is provided at Table 2 and explained in Section 3.5.
>
> ---
>
> **Q1 (c)** Thank you for the question. Providing a full list of aspects of text quality is a difficult question, which is a distinct topic of the research, requiring special attention.

---

### Meta-Review · Area_Chair_tBRC · 2026-01-07

**Summary:**

This paper explores reference-free evaluation of LLM-generated text by analyzing geometric properties of internal representations. The topic is timely and potentially impactful, and reviewers agree that the empirical observation of consistent rankings across models is interesting. However, the reviewers also identify several substantive issues that limit the strength of the current submission.

A primary concern is the conceptual framing of text quality. The paper relies largely on “naturalness” without providing a clear operational definition or explaining why this dimension alone is sufficient. Reviewers question whether the observed correlations reflect general text quality or are driven by properties of the specific datasets and paraphrasing setup.
The theoretical grounding and novelty are also viewed as insufficient. Multiple reviewers note that the paper mainly aggregates existing geometric metrics and reports correlations, without a clearly articulated explanatory mechanism linking representation geometry to text quality. As a result, the contribution is limited.

Reviewers further raise concerns about the limited experimental scope and generalization. The evaluation focuses primarily on movie reviews, making it unclear whether the findings extend to other generation tasks, domains, longer texts, or real-world data.
There are also weaknesses in experimental analysis. Reviewers note the absence of strong baseline comparisons and sensitivity analyses. Reviewers also point to issues of clarity and presentation, particularly in the introduction, which does not clearly state the research question or hypotheses.

Overall, while the paper presents an interesting empirical direction, the reviewers agree that stronger theoretical justification, broader and more rigorous experimentation, and more cautious framing of claims are needed to support acceptance.

**Reviewer Concerns:**

Except Reviewer ZTbS (score 6), I believe other reviewers' concerns are still outstanding. Specifically,

Reviewer wULi's concerns on definition of text quality, generalization beyond paraphrasing, baseline comparisons, and interpretability of metrics, may remain.

Reviewer at29's concerns on novelty and contribution, alternative explanations, statistical rigor, and sensitivity analysis, may remain.

Reviewer 9fFU's concerns on dataset diversity, and real-world generation settings, may remain.

**Reviewer Scores:**

Across reviewers, a full discussion would most likely result in a very small positive score adjustments, driven by improved clarity, acknowledgment of limitations, and partial mitigation of technical concerns. However, many concerns raised by reviewers did not fully addressed based on the rebuttal response. The authors promised to provide many future works, while I believe many of them would still require substantive revisions beyond rebuttal alone.

---

### Decision · Program_Chairs · 2026-01-26

Reject